# Integration of over 9,000 mass spectrometry experiments builds a global map of human protein complexes

Kevin Drew[1] (iD), Chanjae Lee[1,2], Ryan L Huizar[1,2], Fan Tu[1,2], Blake Borgeson[1,2,†], Claire D McWhite[1,2], Yun Ma[2,3], John B Wallingford[1,2] & Edward M Marcotte[1,2,*] (iD)

## Abstract

**Macromolecular protein complexes carry out many of the essential functions of cells, and many genetic diseases arise from disrupting the functions of such complexes. Currently, there is great interest in defining the complete set of human protein complexes, but recent published maps lack comprehensive coverage. Here, through the synthesis of over 9,000 published mass spectrometry experiments, we present hu.MAP, the most comprehensive and accurate human protein complex map to date, containing > 4,600 total complexes, > 7,700 proteins, and > 56,000 unique interactions, including thousands of confident protein interactions not identified by the original publications. hu.MAP accurately recapitulates known complexes withheld from the learning procedure, which was optimized with the aid of a new quantitative metric ($k$-cliques) for comparing sets of sets. The vast majority of complexes in our map are significantly enriched with literature annotations, and the map overall shows improved coverage of many disease-associated proteins, as we describe in detail for ciliopathies. Using hu.MAP, we predicted and experimentally validated candidate ciliopathy disease genes *in vivo* in a model vertebrate, discovering CCDC138, WDR90, and KIAA1328 to be new cilia basal body/centriolar satellite proteins, and identifying ANKRD55 as a novel member of the intraflagellar transport machinery. By offering significant improvements to the accuracy and coverage of human protein complexes, hu.MAP (http://proteincomplexes.org) serves as a valuable resource for better understanding the core cellular functions of human proteins and helping to determine mechanistic foundations of human disease.**

**Keywords** cilia; ciliopathy; human interactome; mass spectrometry; protein complexes; proteomics
**Subject Categories** Genome-Scale & Integrative Biology; Network Biology; Post-translational Modifications, Proteolysis & Proteomics
**Mol Syst Biol. (2017) 13: 932**

## Introduction

A fundamental aim of molecular biology is to understand the relationship between genotype and phenotype of cellular organisms. One major strategy to understand this relationship is to study the physical interactions of the proteins responsible for carrying out the core functions of cells, since interacting proteins tend to be linked to similar phenotypes and genetic diseases. Accurate maps of protein complexes are thus critical to understanding many human diseases (Goh *et al*, 2007; Lage *et al*, 2007; Wang & Marcotte, 2010). Technical advances in the field of proteomics, including large-scale human yeast two-hybrid assays (Rual *et al*, 2005; Rolland *et al*, 2014), affinity purification/mass spectrometry (AP-MS) (Hein *et al*, 2015; Huttlin *et al*, 2015), and co-fractionation/ mass spectrometry (CF-MS) (Havugimana *et al*, 2012; Kristensen *et al*, 2012; Kirkwood *et al*, 2013; Wan *et al*, 2015), have enabled the partial reconstruction of protein interaction networks in humans and other animals, markedly increasing the coverage of protein–protein interactions across the human proteome. Such efforts are largely ongoing, as we still lack a comprehensive map of human complexes, and we have only partial understanding of the composition, formation, and function for the majority of known complexes. Prior high-throughput protein interaction assays in yeast and humans have generally tended to show limited overlap (von Mering *et al*, 2002; Gandhi *et al*, 2006; Hart *et al*, 2006; Yu *et al*, 2008), suggesting that interactions from different studies tend to be incomplete, possibly error-prone, but also orthogonal.

Over the past year, three large-scale mass spectrometry-based protein interaction mapping efforts in particular have greatly expanded the set of known human protein interactions, namely BioPlex (Huttlin *et al*, 2015), Hein *et al* (Hein *et al*, 2015), and Wan *et al* (Wan *et al*, 2015), collectively comprising 9,063 mass spectrometry shotgun proteomics experiments. The three resulting datasets are notable for representing independent surveys of human protein complexes by distinct methods (AP-MS vs. CF-MS),

1 Center for Systems and Synthetic Biology, Institute for Cellular and Molecular Biology, University of Texas at Austin, Austin, TX, USA
2 Department of Molecular Biosciences, University of Texas at Austin, Austin, TX, USA
3 The Otolaryngology Hospital, The First Affiliated Hospital of Sun Yat-sen University, Sun Yat-sen University, Guangzhou, China
*Corresponding author. Tel: +1 512 471 5435; E-mail: marcotte@icmb.utexas.edu
†Present address: Recursion Pharmaceuticals Inc., Salt Lake City, UT, USA

in distinct samples (different cells and tissues), and in the case of the two AP-MS datasets, using distinct choices of affinity-tagged bait proteins. The datasets are complementary in other aspects as well: The two AP-MS interaction sets are each sampled from a single choice of immortalized cancer cell line grown in rich cell culture medium and thus represent deep, but condition- and cell type-specific, views of the interactome network. The AP-MS networks sample only a fraction of human proteins as "baits" and are limited to interactions which contain a bait protein, which is expressed recombinantly as a fusion to an affinity purification moiety (green fluorescent proteins (GFP) for Hein et al or FLAG-HA for BioPlex). These strategies resulted in 23,744 and 26,642 protein interactions for BioPlex and Hein et al, respectively. In contrast, the CF-MS experiments sampled endogenous proteins in their native state without genetic manipulation, but with only partial purification, relying instead on repeat observation of co-eluting proteins across samples and separations to increase confidence in the interactions. The resulting 16,655 protein interactions reflect the biases expected for well-observed proteins, tending toward more abundant, soluble proteins. Additionally, the Wan et al interactome required all interactions to have evidence in at least two sampled metazoan species; thus, only evolutionarily conserved human proteins are represented. As a consequence, none of these three datasets is individually comprehensive; nonetheless, we expect them to present highly complementary, potentially overlapping views of the network of core human protein complexes. There is thus an opportunity to integrate these over 9,000 published mass spectrometry experiments in order to create a single, more comprehensive map of human protein complexes.

Here, we describe our construction of a more accurate and comprehensive global map of human protein complexes by re-analyzing these three large-scale human protein complex mass spectrometry experimental datasets. We built a protein complex discovery pipeline based on supervised and unsupervised machine learning techniques that first generates an integrated protein inter-action network using features from all three input datasets and then employs a sophisticated clustering procedure which optimizes clustering parameters relative to a training set of literature-curated protein complexes. While generating the complex map, we re-analyzed AP-MS datasets to identify > 15,000 high-confidence protein interactions not reported in the original networks. This re-analysis substantially increased the overlap of protein interactions across the datasets and revealed entire complexes not identified by the original analyses. Importantly, the integrated protein interaction network and resulting complexes outperform published networks and complex maps on multiple measures of performance and coverage, and represent the most comprehensive human protein complex map currently available. Moreover, the frame-work we employ can readily incorporate future protein interaction datasets.

We expect that a comprehensive definition of protein complexes will ultimately aid our understanding of disease relations among proteins. In line with expectation, our map shows markedly increased coverage of disease-linked proteins, especially for proteins linked to ciliopathies, a broad spectrum of human diseases characterized by cystic kidneys, obesity, blindness, intellectual disability, and structural birth defects (Hildebrandt et al, 2011).

We highlight both known and novel complexes relevant to ciliopa-thies and, moreover, experimentally validate multiple new protein subunits of ciliary complexes, using in vivo assays of cilia structure and function in vertebrate embryos. Additionally, we distribute our results to the community in a simple and easy to navigate website: http://proteincomplexes.org/. The scale and accuracy of this human protein complex map thus provides avenues for greater understanding of protein function and better disease characterization.

## Results

### Overlap between three recent high-throughput animal protein interaction datasets is modest, but can be greatly increased by a re-analysis of the data

Protein interaction networks from various sources often show mini-mal overlap (von Mering et al, 2002; Gandhi et al, 2006; Hart et al, 2006). We therefore first sought to measure the overlap of proteins and interactions between three recently published protein interac-tion datasets from BioPlex (Huttlin et al, 2015), Hein and colleagues (Hein et al, 2015), and Wan and colleagues (Wan et al, 2015). The BioPlex network is the result of 2,594 AP-MS experiments from HEK293T cells. Similarly, the Hein et al network is the result of 1,125 AP-MS experiments from HeLa cells. In both screens, the authors considered only interactions between the affinity-tagged bait protein and the co-precipitated "prey" proteins, corresponding to a "spoke" model of interactions (Fig 1A). The Wan et al network is derived from a CF-MS analysis of nine organisms, comprising 6,387 MS experiments.

We observe reasonable overlap in terms of the proteins identi-fied within each published network, ranging between 30 and 68% of the proteins between individual networks (Table EV1). However, the overlap among protein interactions was more limited, ranging between ~3 and ~6% overlap (Fig 1B and Table EV1). There are generally three accepted reasons for the limited overlap commonly observed between large-scale protein interaction maps (von Mering et al, 2002): (i) the interaction networks sample different portions of the interactome (e.g., differences in cell types and baits), (ii) the experimental methods used are biased toward discovery of certain classes of interactions (e.g., soluble vs. membrane protein interactions) and therefore are complementary to the other methods, and (iii) the experimental methods produce false-positive interactions.

To further probe the reason for the limited observed overlap, we next considered whether the spoke model interpretation of the AP-MS experiments was partly responsible. By only considering interactions between bait proteins and their preys, spoke models are heavily reliant on the baits selected for experimentation, and also ignore evidence for repeated precipitation of intact complexes across baits. Traditionally, spoke models have shown higher accuracy when compared to the alternative full "matrix" model interpretation (Fig 1C) (Bader & Hogue, 2002). However, the discrimination between true and false protein interactions can be dramatically improved by computing confidence scores for prey–prey interactions when applying a matrix model (Hart et al, 2007; Wang et al, 2009) or a hybrid spoke–matrix model

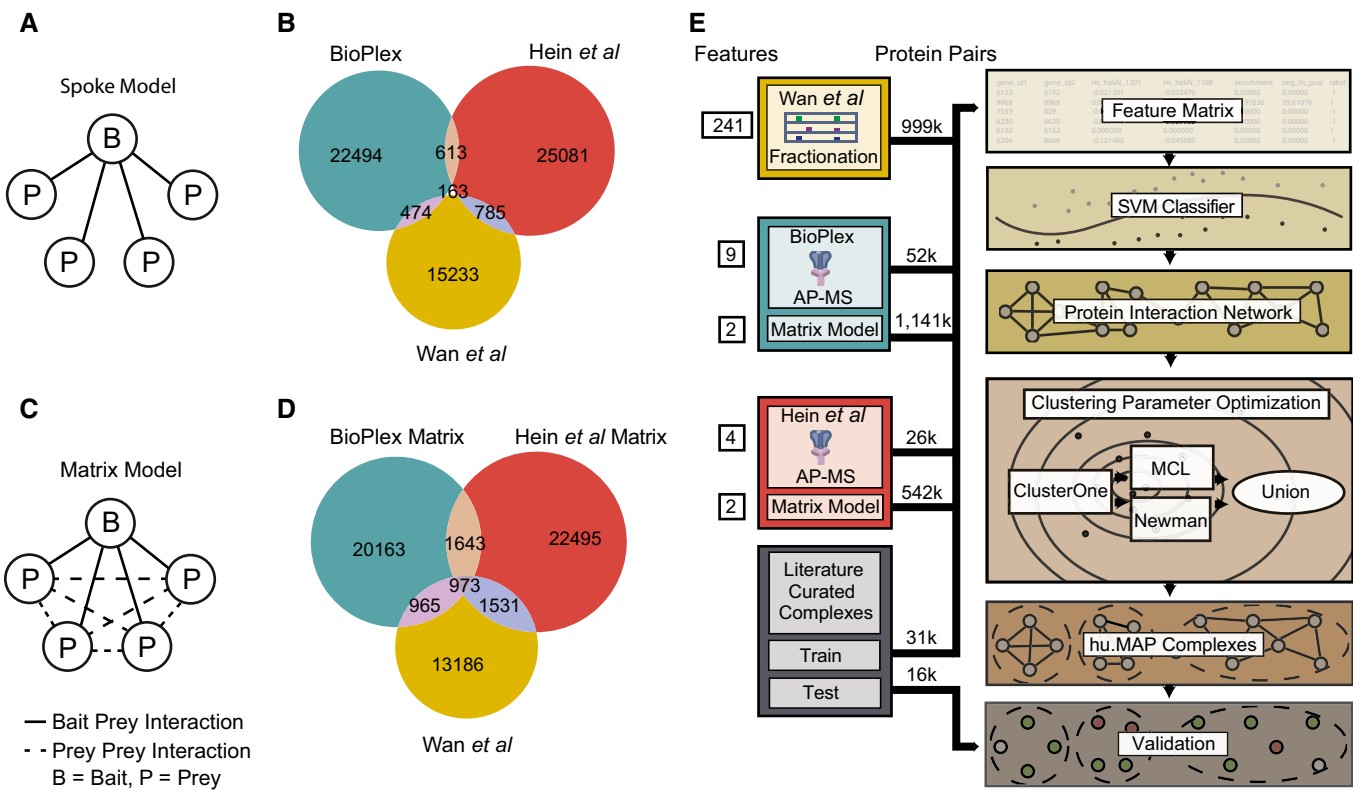

**Figure 1.  Re-analysis of published AP-MS experiments improves overlap among protein interaction networks.**

A   Graphical schematic of spoke model applied to AP-MS datasets. In the spoke model, all interactions must include a bait protein.
B   Venn diagram of overlap between published large-scale protein interaction networks BioPlex (AP-MS), Hein *et al* (AP-MS), and Wan *et al* (CF-MS). Protein interactions in BioPlex and Hein *et al* were generated from a spoke model.
C   Graphical schematic of matrix model applied to AP-MS datasets. In the matrix model, interactions are allowed between prey proteins.
D   Venn diagram of overlap between protein interaction networks where a weighted matrix model was applied to BioPlex and Hein *et al*. Sizes of weighted matrix model protein interaction networks were kept constant with published networks (for this analysis only while the full networks were used for integration). Note an increase in the overall number of overlapping interactions when compared to (B).
E   Diagram of protein complex discovery workflow. Three protein interaction networks, BioPlex, Hein *et al*, and Wan *et al*, were combined into an integrated protein complex network and clustered to identify protein complexes. Parameters for the SVM and clustering algorithms were optimized on a training set of literature-curated complexes and validated on a test set of complexes.

(e.g., socio-affinity index) (Gavin *et al*, 2006) to AP-MS data. In order to reinterpret the AP-MS datasets using a matrix model while effectively discriminating true- and false-positive inter-actions, as well as suppressing "frequent flyer" co-purifying proteins, we applied a hypergeometric distribution-based error model to the AP-MS datasets, calculating *P*-values for pairs of proteins that were significantly co-precipitated more often than random across AP-MS experiments, herein referred to as weighted matrix model. Figure EV1 illustrates a hypothetical example of protein interactions scored using the weighted matrix model and effectively discriminating true from false positives. We then ranked each protein pair according to its calculated *P*-value and selected the top *N* pairs for each AP-MS dataset, where *N* is the number of interactions reported in the original published inter-action networks (23,744 BioPlex interactions and 26,642 Hein *et al* interactions). This reinterpretation of AP-MS experiments using a weighted matrix model substantially increased the amount of overlap among the three interaction networks, which rose to between 10 and 15%, as plotted in Fig 1D (see also Table EV1).

This result indicates that there are thousands of interactions captured by the AP-MS experiments that were not previously identified and confirms a far greater consistency among the underlying mass spectrometry datasets, arguing that a combined analysis of the datasets could considerably improve coverage of the complete human protein interactome.

## Integrating the large-scale proteomics datasets into a human protein–protein interaction network

Based on the notion that considering this large and diverse set of experiments jointly should increase the ability to discriminate between true and false protein interactions, we next asked whether integrating all three large-scale datasets would outperform the individual networks in terms of identifying true human protein interactions. We employed a formal machine learning framework to combine evidence from the thousands of individual mass spec-trometry experiments in the three large-scale datasets. Our approach was specifically designed to address the limited network

overlap described above, using the weighted matrix model to increase interaction coverage while preserving accuracy. We expected the orthogonal techniques employed, CF-MS and AP-MS, to complement each other, where CF-MS captures stable interactions among endogenous proteins in diverse cells and tissues, while AP-MS captures a large collection of interactions with differing biophysical characteristics. The three datasets also sample very different portions of the human interactome in terms of cell type and bait selection, which we similarly expected to contribute to a more comprehensive map.

Figure 1E outlines the pipeline used for protein complex discovery. We first generated a feature matrix using the raw untrained published features from BioPlex, Hein *et al,* and Wan *et al* as well as the new weighted matrix model features, in the form of a negative log hypergeometric *P*-value capturing the specificity and extent to which pairs of proteins co-precipitated across many AP-MS baits. Rows in the feature matrix represented pairs of proteins and columns represented measured numerical estimates of protein pairs' interaction potentials based on the different experiments. All protein interaction features were calculated from raw experimental data, and to avoid any circularity, no features trained on our gold standard were used (see Materials and Methods). We also labeled protein pairs according to their support by a gold standard, literature-curated set of human protein complexes [the CORUM protein complex database (Ruepp *et al*, 2010)]. We assigned a positive label if both proteins were seen in the same complex, a negative label if both proteins were observed in the literature-curated set but not in the same complex, and an "unknown" label for all other pairs. A support vector machine (SVM) classifier was trained using the labeled feature matrix, then applied to all protein pairs, assigning each pair an SVM confidence score, indicating the level of support for that pair of proteins to participate in the same complex. This classification step thus resulted in an integrated human protein–protein interaction network, in which the nodes are proteins identified in any of the three experimental datasets, and the edges between nodes represent co-complex interactions weighted proportionally to the SVM score.

As an initial estimate of the quality of the integrated human protein interactions, we calculated their precision and recall by reconstructing a set of 15,687 gold standard, literature-curated co-complex interactions omitted from the training procedure. While networks generated using features from only one of the three datasets showed high precision for high-confidence interactions, they quickly dropped in precision in the higher recall range (Fig 2A). In contrast, the integrated network demonstrated substantial improvements to performance, with a precision of 80% over just under half of the benchmark interactions. Additionally, adding the weighted matrix model features to the published interactions greatly improved the performance, indicating that the weighted matrix model features capture new information beyond spoke features and serve as a rich source of evidence supporting true protein interactions (Figs 2A, and EV2A and B).

Previous studies using proteomics data for interaction identification saw gains in performance when non-physical data (co-expression, co-citation, etc.) were included in training. Specifically, Wan *et al* (2015) included HumanNet (Lee *et al*, 2011) features (only for protein pairs when there was also evidence in the co-fractionation data), which showed a boost in performance.

Since we used features from Wan *et al*, we wanted to test the value of the non-physical data in our pipeline. Figure EV2C shows precision–recall curves for interaction networks trained without literature-based evidence from HumanNet as well as a network trained without all of HumanNet. Negligible performance loss is observed when HumanNet is removed, suggesting large-scale human protein interaction datasets have reached a sufficient point where adding in supporting non-physical interaction information is no longer necessary to support protein interaction discovery.

## Clustering pairwise interactions reveals human protein complexes

A hallmark of protein complexes is that their component proteins should frequently be co-purified in independent separations and affinity purifications. This trend manifests as densely connected regions of the interaction network, which we sought to identify by applying a two-stage clustering procedure. In the first stage of clustering, we applied the ClusterOne algorithm (Nepusz *et al*, 2012), which identifies large, dense sub-networks of the full protein interaction network. Importantly, ClusterOne allows proteins to participate in more than one sub-network as dictated by the data, as proteins frequently participate in more than one complex (Wan *et al*, 2015). In the second stage, we separately applied MCL (Enright *et al*, 2002) and Newman's hierarchical clustering method (Newman, 2004) to further refine the sub-networks produced by ClusterOne. As with many unsupervised machine learning techniques, clustering algorithms have adjustable parameters for optimizing their performance. We therefore used a parameter sweep strategy to identify choices of parameters that best recapitulated known complexes. We evaluated each parameter combination by comparing the resulting protein clusters to our literature-curated training set of protein complexes and selected the top-ranking parameter combination. As the comparison of protein complexes to a gold standard set is not a fully solved problem, we first developed an objective scoring framework for complex-level precision and accuracy, called *k*-cliques as we describe in the Materials and Methods. This method allows us to compare predicted sets of complexes to a gold standard to evaluate their similarity on a global level.

We computed the performance in terms of reconstructing known complexes for each of > 1,000 different clustering algorithm parameter combinations, varying the SVM confidence threshold for the input pairwise protein interactions, the ClusterOne density and overlap options, and the inflation option for MCL. The top-scoring sets of clusters for the two second-stage clustering methods, MCL and Newman's hierarchical method, were of similarly high quality when evaluated relative to the training set of complexes (Fig 2B). These two top-scoring cluster sets also showed the top-ranking scores when compared to the literature-curated leave-out test set for their respective clustering methods, serving to validate the parameter optimization method. As the two top-scoring cluster sets identified many distinct specific complexes and sub-complexes, we combined these two top-scoring definitions of complexes in order to provide a more comprehensive view of the myriad of physical protein assemblies in human cells. The resulting fully integrated human protein complex map, called hu.MAP, consists of 4,659 complexes, 56,735

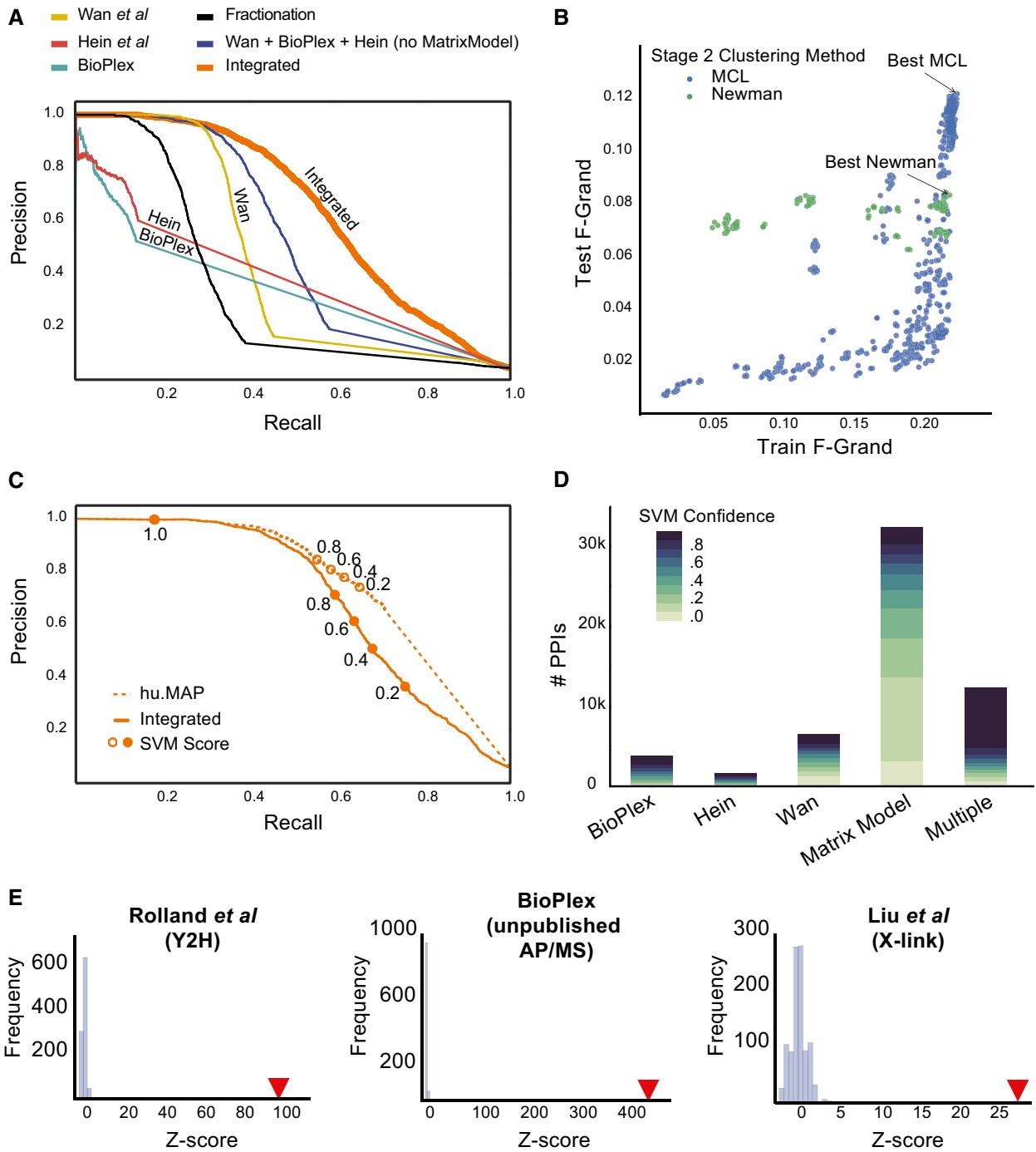

**Figure 2.  Integration of the three large-scale protein complex datasets substantially improves both precision and recall of known human protein interactions.**

A   Precision–recall curves calculated on a leave-out set of protein interactions from literature-curated complexes for different combinations of predictive protein interaction features. The integration of all three datasets outperforms all other networks. Also, note a substantial improvement in performance when the weighted matrix model features are used (no MatrixModel, blue vs. integrated, orange).

B   Performance of parameter optimization for MCL and Newman two-stage clustering procedures. Each data point represents a set of parameters and is evaluated based on the resulting clusters similarity to both training and test sets of complexes using the *F*-Grand measure (see Materials and Methods). Final parameter sets were selected based only on *F*-Grand measure for the training set.

C   Precision–recall curves evaluating protein interactions on leave-out set before (integrated) and after (hu.MAP) clustering procedure. Note an improvement in performance after clustering suggests the clustering procedure successfully removed false-positive interactions.

D   Distribution of protein interactions in the final protein interaction network based on input evidence. Note the weighted matrix model interactions produce many high-confident interactions. Also, the "Multiple" category shows predominately high-confident interactions, which validates the integration of multiple datasets mitigating false positives.

E   Protein interactions from our complex map substantially overlap with other protein interaction datasets across a variety of experimental types.

unique co-complex interactions and 7,777 unique proteins (Tables EV2 and EV3).

## The integrated map improves pairwise interaction performance, identifies new interactions, and is strongly supported by independent protein interaction datasets

We wished to assess the quality of the integrated map of human protein complexes by multiple, independent approaches. First, because the process of network clustering entails removing interactions between proteins that are inconsistent with the defined complexes, we might expect the resulting clustered network to be more accurate than the pre-clustered network. Indeed, the final interaction network shows improved precision and recall (Fig 2C), indicating that the clustering step is preferentially removing false positives from the original network.

Next, during the course of identifying protein interactions and complexes, we withheld a leave-out set of literature-curated complexes to serve as a final, fully independent test set. We compared these data to the derived map and to previously published complex maps, using two different comparison measures (Fig EV2D and E). For both the $k$-clique metric and the precision–recall product measure (Song & Singh, 2009), we observed a dramatic improvement in performance over the Wan *et al* and BioPlex maps (note: Hein reported only interactions, not complexes). We also observe in Fig EV2F a broad distribution of complexes with various numbers of subunits in our map, with 2,991 (64% of the total) having greater than two subunits, suggesting that our clustering procedure is capable of identifying the full range of complex sizes. A survey of evidence supporting each interaction in the map showed multiple lines of evidence supported many pairwise interactions (Fig 2D). This further supports the notion that the underlying datasets are orthogonal and that integrating them provides substantial improvement on discriminating true and false protein interactions. Remarkably, however, we observed tens of thousands of interactions in the map supported only by weighted matrix model features, 15,454 of them having very high confidence (SVM score > ~0.27, see Materials and Methods). Thus, considering prey–prey interactions in the AP-MS datasets dramatically enhanced the identification of human protein interactions.

Finally, in order to assess the quality of the final map independently of both the test and training set complexes, we further evaluated our complex map with several of the largest remaining available human protein interaction datasets. We observed highly significant overlap with protein interactions from different experimental methods, including yeast two-hybrid assays (Rolland *et al*, 2014), additional unpublished BioPlex AP-MS experiments (BioPlex), and cross-linking mass spectrometry performed on human cell lysate (Liu *et al*, 2015) (Fig 2E). Thus, comparisons with independent datasets strongly support the high quality of the derived protein complexes, as measured by multiple metrics of performance, considering interactions both pairwise and setwise, and even considering interactions measured independently by multiple different technologies. The significant overlap of our complex map with these other datasets also points toward the potential value of integrating these datasets using the pipeline described here to further improve coverage of the human protein interactome.

## Prey–prey interactions reveal a large, synaptic bouton complex, isolated from HEK cells

The thousands of additional high-confidence interactions contributed by prey–prey co-purification patterns led us next to consider their value in our protein complex discovery pipeline. In particular, we asked whether weighted matrix model edges could independently identify complexes, or whether they only served to support observed bait–prey associations. We thus searched for complexes in the map that were supported predominantly with weighted matrix model interactions. Figure 3A summarizes AP-MS experiments for four example complexes. Three of these complexes—the exosome complex, eukaryotic initiation factor 3 (eIF3) complex, and the 19S proteasome—were supported by both spoke edges and weighted matrix model edges, showing high complementarity between the two sets of interactions. This support was evident in the strong interaction density both between bait proteins and between bait and prey proteins within each complex. In contrast, the fourth complex shown in Fig 3A is a newly identified complex by our pipeline that surprisingly has limited density between bait proteins, but substantial, high-specificity density in the prey region of the matrix. Notably, the four bait proteins that each precipitates nearly all 60 subunits of this complex largely do not co-precipitate each other.

We performed annotation enrichment analysis to establish functional connections between member proteins of this novel complex. Strikingly, the proteins identified in this complex are highly specific for cerebral cortex tissue, as measured by Human Protein Atlas tissue expression data (Uhlén *et al*, 2015) (Fig 3B). We additionally observed high brain-region-specific co-expression among members of the complex, unlike as for random protein pairs, in the Allen Brain Map microarray dataset (Hawrylycz *et al*, 2012) (Fig 3C). The complex includes subunits of the SNARE complex, a known physically associated set of proteins involved in synaptic vesicles (Südhof, 1995). Consistent with this trend, we found a strong enrichment of Gene Ontology terms (GO) (Ashburner *et al*, 2000) among members of the complex specific to neurotransmission and neuron migration (Fig 3D). Thus, there is good correspondence between this complex and known interacting protein complexes at the synaptic bouton, the presynaptic axon terminal region containing synaptic vesicles, and the location of neuronal connections.

We next wanted to consider the possibility that the set of synaptic bouton complex proteins were co-precipitating due to their enclosure in a membrane-bound organelle rather than making physical co-complex interactions. We therefore looked at the expected number of proteins we would find given the purification of a synaptic vesicle. Out of the 131 proteins annotated with the GO term *Synaptic Vesicle* (GO:0008021), 66 are found in complexes in hu.MAP, but only 12 are found in the synaptic bouton complex. This low recovery of synaptic vesicle proteins suggests that the synaptic bouton complex is not exclusively vesicle-bound components.

Rather surprisingly, the AP-MS experiments that support this complex were all performed with HEK293T cells. HEK293T cells were first reported to be derived from human embryonic kidney tissue (Graham *et al*, 1977), and therefore, it is puzzling as to why a complex comprised of cerebral cortex-specific proteins showed such a strong signal in kidney-derived cells. However, re-analyses

of HEK293T cell origins suggest that they were originally mis-annotated and actually derive from adjacent human embryonic adrenal tissue, rather than embryonic kidney cells (Shaw *et al*, 2002; Lin *et al*, 2014), and thus exhibit many neuronal properties (Shaw *et al*, 2002). The possibility remains open that the protein complex identified here could also have additional roles in the body. Nevertheless, this complex exemplifies the value of prey–prey interactions for discovering protein complexes.

## The integrated map markedly improves coverage of disease-linked protein complexes

A key application of more accurate human protein complex maps will be to highlight and characterize biologically important protein modules, especially those relevant to human disease. We thus next evaluated the map in reference to a variety of localization, functional, and disease annotation datasets. First, we annotated proteins in hu.MAP with information about their human tissue expression patterns from the Human Protein Atlas (Uhlén *et al*, 2015). We observed a substantial portion of proteins in our map expressed across all assayed tissues, suggesting our map captures many core processes in human cells (Fig 4A), although many tissue-specific complexes appear to be identified as well, as for the example of the synaptic bouton complex in Fig 3.

We next evaluated the fraction of complexes with significantly enriched annotations [FDR-corrected hypergeometric test; g:Profiler (Reimand *et al*, 2016)] from the Gene Ontology, Reactome, CORUM, OMIM, KEGG, and HPA annotation databases (Ashburner *et al*, 2000; Ruepp *et al*, 2010; Kanehisa *et al*, 2014; Amberger *et al*, 2015; Uhlén *et al*, 2015; Fabregat *et al*, 2016). While nearly all of the complexes in hu.MAP (4646/4659) have at least one significantly enriched annotation when searched individually (see Table EV4 for full list of each complexes' significantly enriched annotation terms), in order to better estimate annotation enrichments considering the > 4,000 distinct complexes being tested and the non-uniform complex size distribution (Fig EV2F), we additionally estimated the occurrence of significant enrichment by chance after permuting protein memberships in complexes while maintaining the observed distribution of complex sizes. Figure EV3 shows the distribution of the largest -log(*P*-values) (i.e., most significant annotation) for each complex for both hu.MAP and the shuffled complexes. Figure 4B reports the set of hu.MAP complexes with a significantly enriched annotation at a false discovery rate of 5% with respect to the shuffled set of complexes. Greater than 40% (1,880 out of 4,659) of the complexes had at least one significantly enriched annotation term, demonstrating the biological pertinence of complexes in the map, well in excess of shuffled complexes of the same sizes. While many complexes of size 2 and size 3 were included in the set of 1,880 annotation-enriched complexes, larger complexes were increasingly more likely to show functional enrichment, with 1,514 enriched complexes containing three or more subunits.

Knowledge that a protein interacts with a disease-associated protein greatly increases the probability that the first protein is linked to the same disease (Dudley *et al*, 2005; Fraser & Plotkin, 2007; Lage *et al*, 2007; McGary *et al*, 2007; Ideker & Sharan, 2008). Thus, we expect an important application of this map will be to enable the discovery of candidate disease genes. In order to estimate this strategy's potential, we compared the map's coverage of known

disease-associated proteins with other published networks. Figure 4C shows the fraction of proteins annotated in the Online Mendelian Inheritance in Man (OMIM) disease gene database, mapped according to eight high-level Disease Ontology (DO) terms (Schriml *et al*, 2012) for four complex maps [i.e., hu.MAP, Wan *et al*, BioPlex, and a targeted cilia map from Boldt *et al* (2016)]. We also evaluated three interaction networks (which serve to increase its proteome coverage) specifically the full Hein *et al* interaction network and the two targeted interaction networks of Gupta *et al* (2015) (centrosomal) and Boldt *et al* (cilia). hu.MAP shows substantially higher coverage than the other networks for nearly all high-level DO terms, covering ~46% of the annotated human disease-associated proteins.

## New components of ciliary protein complexes

One specific class of diseases in particular stood out, namely diseases related to defective cilia, known as ciliopathies. Cilia are microtubule-based cellular protrusions that are critical for cell-to-cell signaling (Eggenschwiler & Anderson, 2007; Oh & Katsanis, 2012) and proper embryonic development (Goetz & Anderson, 2010; Oh & Katsanis, 2012). Cilia assembly and maintenance are highly regulated processes whose disruption can lead to debilitating birth and early childhood disorders, including Joubert syndrome, Meckel syndrome, Bardet–Biedl syndrome, orofaciodigital syndrome, and polycystic kidney disease. Although many ciliopathies share clinical presentations such as kidney and liver dysfunction, other clinical features and their severity can vary considerably across individuals (Gerdes *et al*, 2009; Tobin & Beales, 2009; Hildebrandt *et al*, 2011). The resulting confounding array of clinical features, an absence of cures, and limited but expensive treatments all lead ciliopathies to collectively represent a major health burden (Tobin & Beales, 2009). Protein complexes are integral to many ciliary and centrosomal processes and have major roles in ciliopathies (Gupta *et al*, 2015; Boldt *et al*, 2016). To more directly assess hu.MAP's relevance to ciliopathies, we measured its coverage of ciliopathy-associated proteins (OMIM-annotated proteins mapped onto the mid-level Disease Ontology term "ciliopathy") and known ciliary proteins [literature-curated as the SysCilia "Gold Standard" (van Dam *et al*, 2013)] (Fig 4C). For both ciliopathy-associated and ciliary proteins, we observed a substantial increase in coverage over other general networks and complex maps, with hu.MAP covering > 50% of ciliary proteins. Additionally, when compared directly to the more targeted networks and complex map of Gupta *et al* and Boldt *et al*, our map is comparable with regard to "ciliopathy" proteins and exceeds coverage with regard to SysCilia proteins (Fig 4C). Due to our map being proteome-wide, we anticipate similar levels of performance across many other specific disease types.

An examination of individual complexes enriched with ciliary proteins highlighted both known and novel ciliary components. hu.MAP reconstructed multiple known ciliary protein complexes including the Intraflagellar Transport particles A and B (IFT-A and IFT-B) (Piperno & Mead, 1997; Cole *et al*, 1998), the Bardet–Biedl-linked BBSome (Nachury *et al*, 2007), the CPLANE ciliogenesis and planar polarity effector complex (Toriyama *et al*, 2016), and the CEP290-CP110 complex (Tsang *et al*, 2008) (Fig 5). In all, the map contains 234 complexes and sub-complexes involving 158 ciliary proteins (Table EV5), many associated with ciliopathies (den

Hollander *et al*, 2006; Beales *et al*, 2007; Chetty-John *et al*, 2010; Walczak-Sztulpa *et al*, 2010; Schaefer *et al*, 2014; Toriyama *et al*, 2016). Moreover, we observed many of these complexes to also contain additional uncharacterized proteins. These novel proteins represent excellent candidates for ciliary roles including potential links to ciliopathies and points to the broad use of hu.MAP for associating uncharacterized genes to disease phenotypes based on co-complex interactions. We therefore next focused on detailed experiments to characterize novel proteins' *in vivo* functions and subcellular localization in developing vertebrate embryos.

### Observation of an 18-subunit ciliopathy-linked complex enriched in centrosomal proteins

Among the ciliary complexes, we identified a large, 18-subunit complex in which eight subunits were already linked to ciliopathies and 14 members were known to localize to the centrosome centriolar satellites (Figs 5 and 6A). A second 8-member complex was observed interacting with subunits of the first complex, also containing centrosome-localized and ciliopathy-linked proteins (Fig 6A). Figure 6B plots the AP-MS observations that supported the discovery of these complexes. We observed strong evidence for physical associations among members in each complex, with many edges supported by our weighted matrix model as well as affinity purification of substantial portions of each complex by bait proteins from multiple datasets. Centrosomes are the microtubule organizing centers of cells, with dual roles in chromosomal movement and organization of the ciliary microtubule axonemes. Thus, the marked enrichment of centrosomal/centriolar satellite and ciliopathy proteins in these two complexes strongly implicates a relationship between centriolar satellites and ciliary-related disease.

Three of the 18 proteins in the larger complex were completely uncharacterized (WDR90, CCDC138 and KIAA1328), so we determined the subcellular localization of tagged versions of these proteins as a direct experimental test of the map's prediction. We expressed proteins in this complex as GFP fusions in multi-ciliated cells (MCCs) of embryos of the frog *Xenopus laevis*, as these cells provide an exceptional platform for studying vertebrate ciliary cell biology *in vivo* (Brooks & Wallingford, 2012; Werner & Mitchell, 2012; Toriyama *et al*, 2016). Serving as positive controls, known centrosomal components, including PIBF1, localized strongly and specifically to basal bodies and co-localized with the basal body marker Centrin4 (Fig 6C). WDR90, CCDC138, and KIAA1328 each localized strongly and specifically to basal bodies, strongly supporting their participation in centrosomal and ciliary biology, and validating the map's predictions.

### ANKRD55 is a novel intraflagellar transport complex protein

We next focused on the IFT complexes, which link cargos to microtubule motors for transport along ciliary axonemes (Taschner & Lorentzen, 2016). The IFT system is comprised of two multi-protein complexes, IFT-A and IFT-B (Piperno & Mead, 1997; Cole *et al*, 1998). Our map effectively recapitulated known protein–protein interactions in the IFT-B complex, assembling not only the entire complex (Figs 5 and EV4A), but also recovering elements of known sub-complexes. For example, the map assembled much of the known IFT-B "core" (also called the IFT-B1 complex) containing IFT22, IFT46, IFT74, and IFT81. The map also identified a complex containing IFT38, IFT54, IFT57, and IFT172, which closely matches the recently described IFT-B2 complex (Katoh *et al*, 2016; Taschner *et al*, 2016). The map further recapitulated the smaller IFT-A complex (Figs 5 and EV4A), the anterograde IFT motor complex of KIF3A, KIF3B, and KAP (Taschner & Lorentzen, 2016), and also more ancillary but relevant interactions, such as that between IFT46 and the small GTPase ARL13B (Cevik *et al*, 2013).

Importantly, the map also predicted novel components of the IFT complexes. For example, the map predicted an interaction between IFT-B and RABEP2 (Fig 5), which is interesting because while RABEP2 is implicated in ciliogenesis (Airik *et al*, 2016), its mechanism of action remains obscure. Even more interesting is the link between IFT-B and the poorly defined protein ANKRD55 (Figs 5 and 7A). A re-examination of the raw data from the AP-MS experiments reinforced the notion that ANKRD55 is an IFT-B component (Fig 7B), with the evidence coming entirely from BioPlex. Interestingly, ANKRD55 is not present in complexes in the BioPlex map but is identified in hu.MAP, suggesting our clustering procedure is highly sensitive to previously overlooked complexes. We then tested the hypothesis that ANKRD55 is an IFT-B component *in vivo* using high-speed confocal imaging in *Xenopus* MCCs (Brooks & Wallingford, 2012). We find that an ANKRD55-GFP fusion protein localizes to cilia, and moreover, time-lapse video analysis indicates that ANKRD55 traffics up and down cilia (Fig 7C and Movie EV1). In kymographs made from the time-lapse data, we observed ANKRD55-GFP to move coordinately in axonemes with known IFT protein CLUAP1-RFP (Fig EV5A, and Movies EV2 and EV3). Finally, disruption of the ciliopathy protein JBTS17 was recently shown to elicit accumulation of IFT-B proteins (but not IFT-A proteins) in ciliary axonemes (Toriyama *et al*, 2016). Consistent with the predicted association of ANKRD55 with IFT-B, we observed robust aberrant accumulation of ANKRD55 in axonemes after *JBTS17* knockdown (Fig EV5C).

---

**Figure 3.  Weighted matrix model edges identify large synaptic bouton complex.**

A   Presence/absence matrix of BioPlex AP-MS experiments as rows and pulled down proteins as columns for four complexes identified in our complex map. The Exosome, eIF3 Complex, and 19S Proteasome all have multiple bait–bait interactions whereas the novel synaptic bouton complex does not have bait–bait interactions but does have substantial density in the non-bait region of the matrix. This density is identified by the weighted matrix model and highlights the model's ability to discover protein complexes.

B   RNA expression profiles of proteins in the synaptic bouton complex across different tissues sampled by the Human Protein Atlas. This shows the complex is highly specific for cerebral cortex tissue. No less than six replicates were used for each tissue type. Boxes indicate median (inner band), first quartile (bottom) and third (top) quartile. Whiskers indicate 1.5 interquartile range. Dots indicate outliers.

C   Correlation coefficient distributions of Allen Brain Map tissue expression profiles between synaptic bouton complex proteins and random set of proteins. This shows coherence of expression among proteins in the complex suggesting a functional module.

D   Significantly enriched Gene Ontology annotations for proteins in the synaptic bouton complex shows enrichment for neuron development and synaptic transmission.

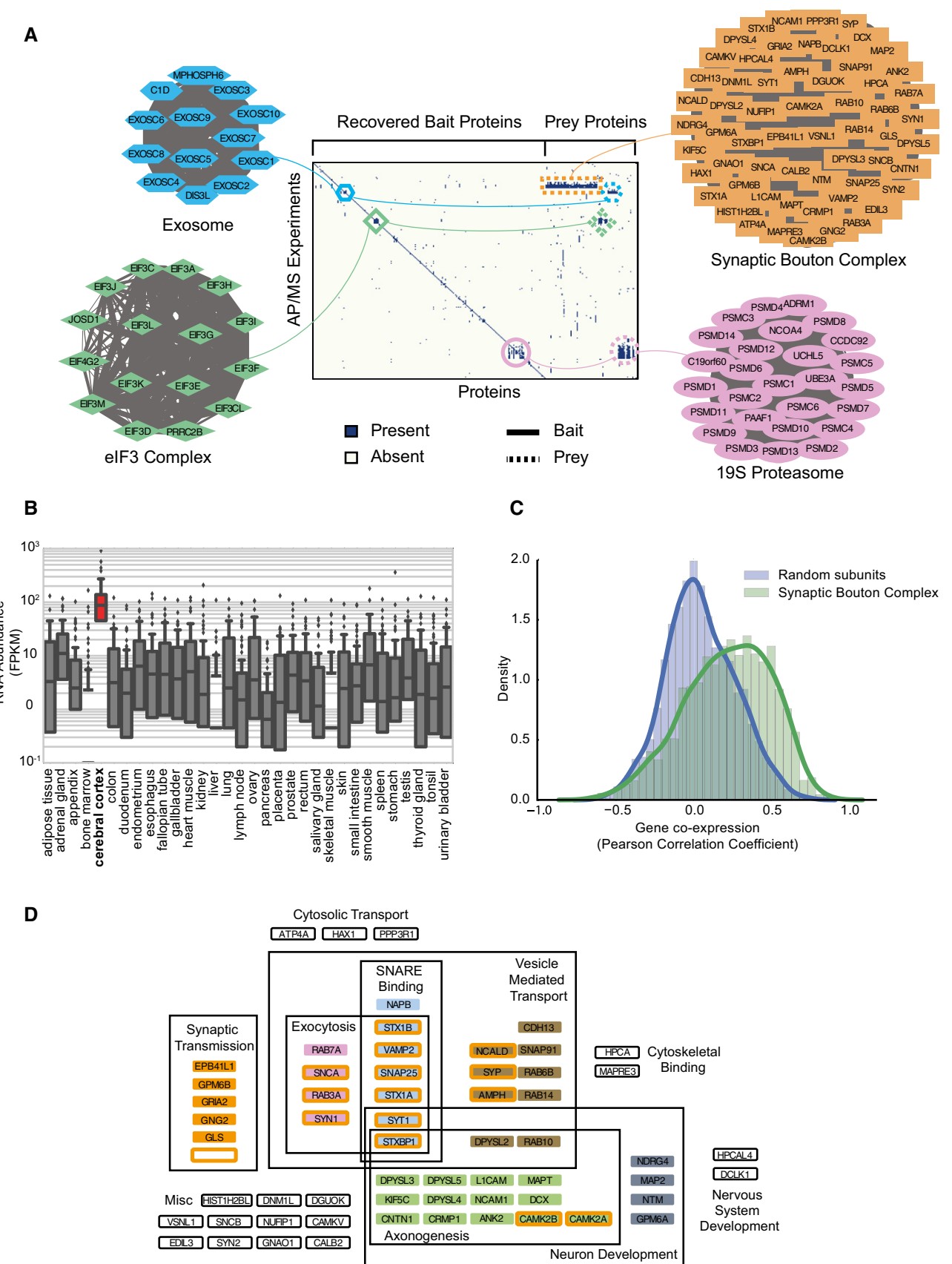

**Figure 3.**

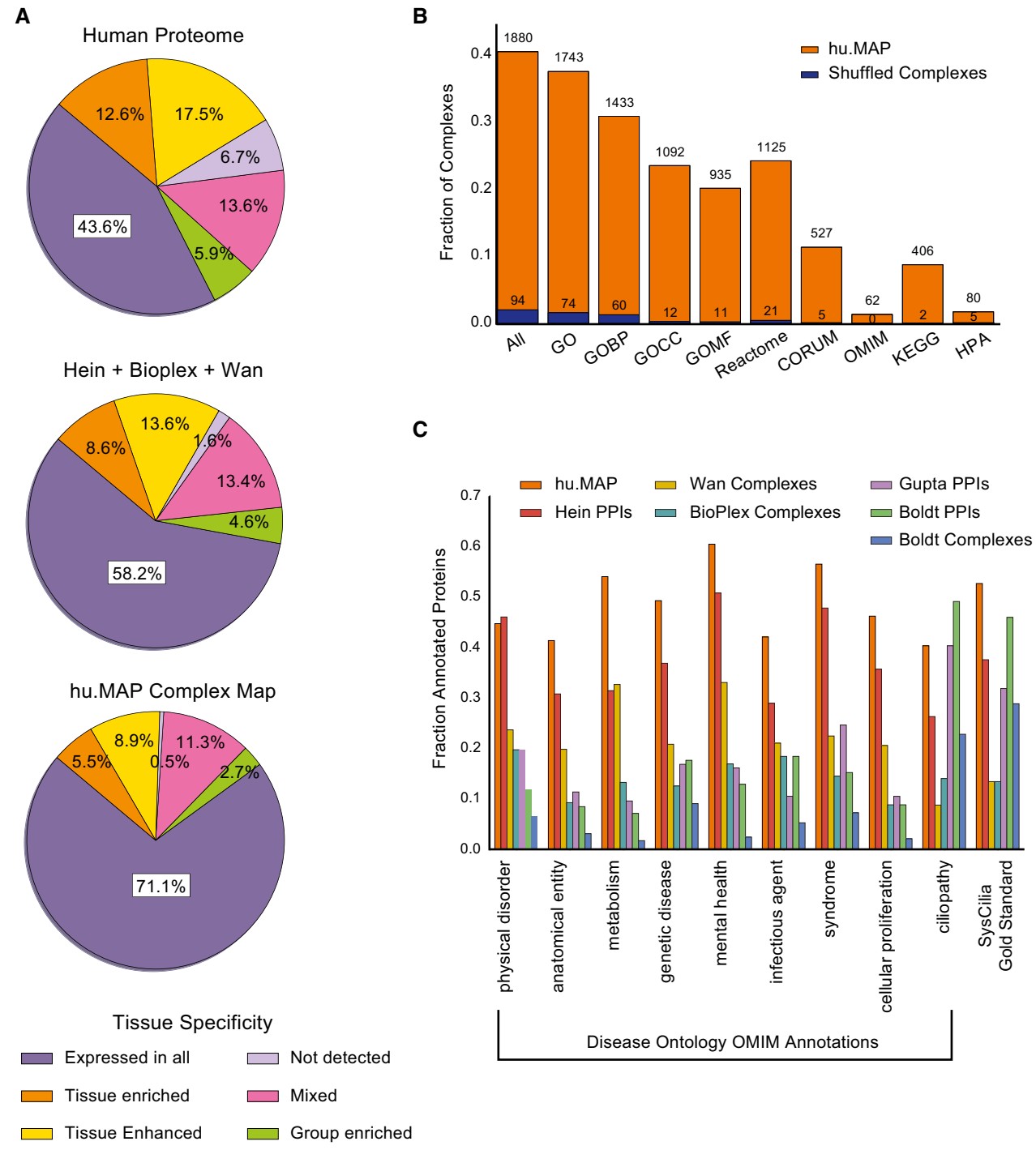

**Figure 4.  hu.MAP consists of predominately core human complexes and covers a large fraction of disease genes.**

A   Complex map coverage of Human Protein Atlas RNA tissue specificity classifications showing majority of complexes are ubiquitously expressed and likely core cellular machinery.

B   Fraction of complexes with significantly enriched annotation terms (g:Profiler hypergeometric test with FDR (Benjamini–Hochberg) correction on each complex and further corrected at an FDR of 5% given a set of shuffled complexes; see Materials and Methods) from various ontologies.

C   Protein coverage of high-level Disease Ontology terms and cilia-related annotations for complex map as well as three published maps (Wan *et al*, BioPlex, Boldt *et al*) and three published interaction network (Hein *et al*, Gupta *et al* and Boldt *et al*).

Because IFT subunits have been linked to vertebrate birth defects, as a new subunit of the IFT-B particle, we would expect disruption of ANKRD55 to in turn disrupt ciliary function and

proper embryonic development. We performed *in vivo* experiments in order to test ANKRD55 function, first by asking whether knockdown elicited a similar cilia phenotype to IFT knockdown. Fig 7D

shows images of morpholino antisense oligonucleotide (MO) knockdowns of both *ANKRD55* and its co-complex member *IFT52*; the morphant embryos exhibit similar ciliary disruption phenotypes, further supporting the connection between IFT and ANKRD55. Finally, disruption of IFT in vertebrate animals, including *Xenopus* and mice, results in defects in neural tube closure (Huangfu *et al*, 2003; Toriyama *et al*, 2016), and we therefore asked whether loss of ANKRD55 would exhibit similar defects. Indeed, knockdown of

*ANKRD55* in *Xenopus* embryos resulted in defective neural tube closure; this defect could be rescued by expression of a version of the *ANKRD55* mRNA that could not be targeted by the MO, arguing for the specificity of the knockdown phenotype (Fig 7E). Taken together, the interaction, localization, and genetic perturbation data all indicate that ANKRD55 interacts physically and functionally with the IFT-B complex and strongly suggests that ANKRD55 is likely to play a role in human ciliopathies.

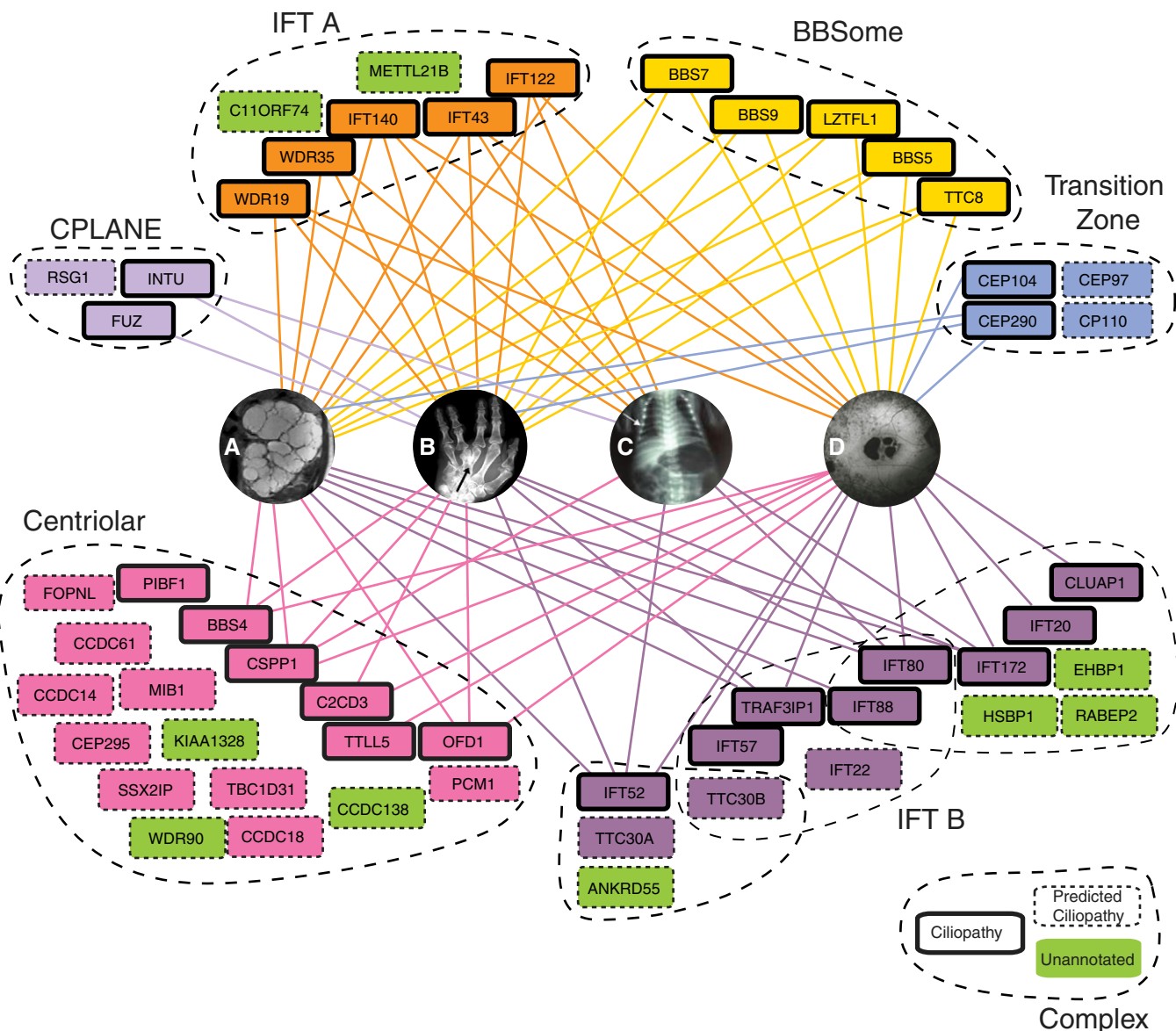

**Figure 5.  Select complexes in the map are strongly linked to human ciliopathies.**
Eight complexes are highlighted with ciliopathy-linked subunits (bold outlines), predicted ciliopathy subunits (dashed outlines), and their association with four representative ciliopathy phenotypes (A–D). We predict links to ciliopathies for uncharacterized proteins (green) that are co-complex with known ciliopathy genes. All edges to ciliopathy phenotypes are mapped from OMIM (Amberger *et al*, 2015) or direct from literature (Krock & Perkins, 2008; Keady *et al*, 2011; Chang *et al*, 2015; Toriyama *et al*, 2016).

A    Cystic kidney phenotype represented by polycystic kidneys from patient with OFD1 variant, adapted from Chetty-John *et al* (2010).
B    Digit malformations represented by polydactyly of Bardet–Biedl syndrome patient with LZTFL1 (BBS17) variant, adapted from Schaefer *et al* (2014).
C    Short-rib phenotype represented by chest narrowing of Jeune asphyxiating thoracic dystrophy individual with IFT80 variant, adapted from Beales *et al* (2007).
D    Maculopathy represented by retinitis pigmentosa of Senior–Loken syndrome patient with mutation in WDR19 (Coussa *et al*, 2013).

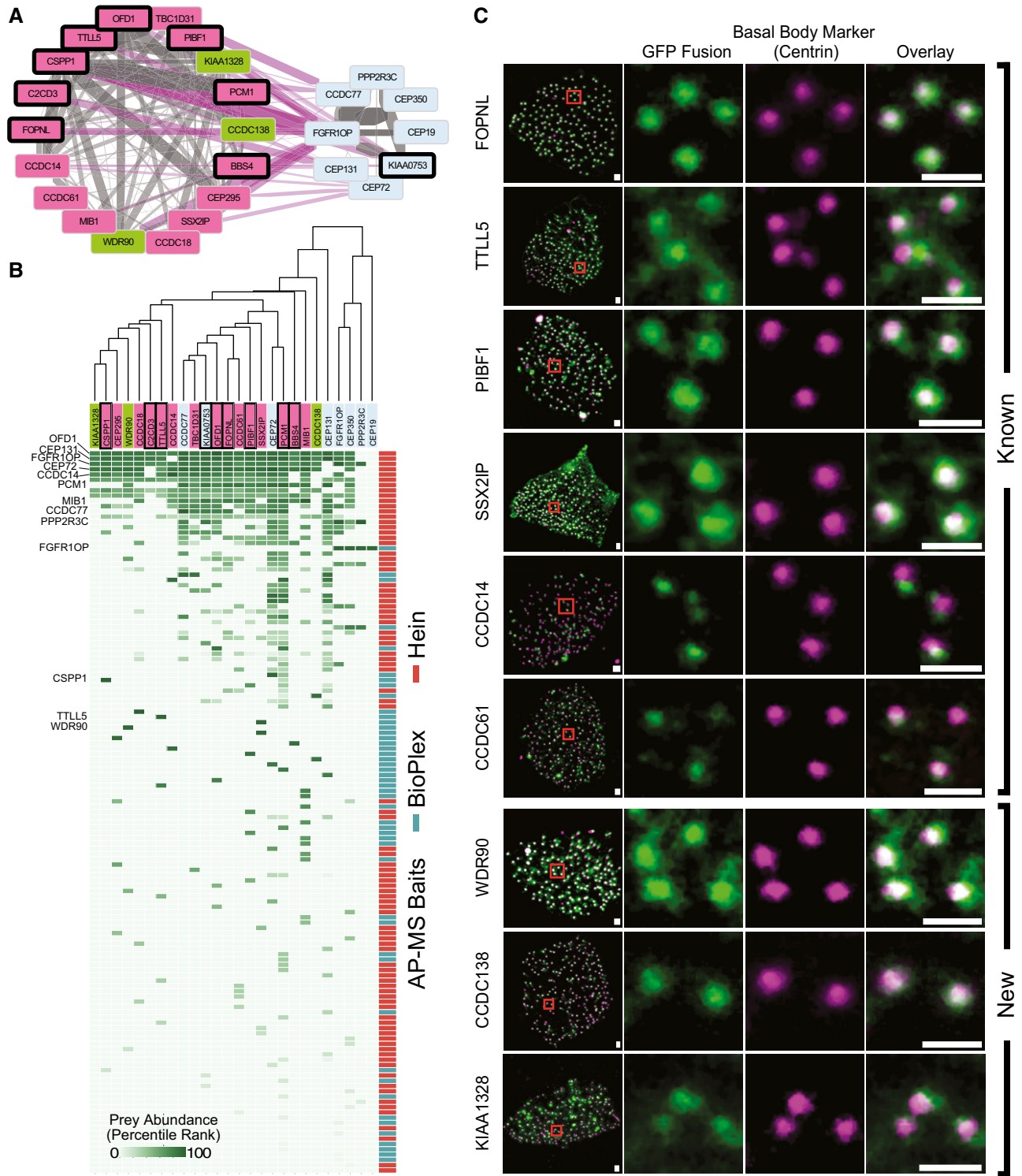

**Figure 6.  Oro-facial-digital syndrome 1 (OFD1) interaction partners are centriole and centriolar satellite proteins, suggesting new components of ciliary basal bodies.**

A   Network of ciliopathy complex and closely interacting centrosomal complex. Edge weights represent SVM confidence scores where gray are intracomplex edges and purple are inter-complex edges. Color of nodes follows Fig 5 conventions.

B   Matrix of AP-MS evidence supporting both complexes. The matrix shows strong support for interactions within each complex. Bait proteins that are members of either complex are labeled on the left.

C   Experimental validation of ciliary proteins using multi-ciliated epithelial cells in *Xenopus laevis*. Localization assays for the three uncharacterized proteins in the OFD1 complex confirm that all three proteins localize to basal bodies at the base of the cilia in a manner similar to known components of the complex. Scale bars: 1 μm. Each image is representative of nine cells from three different embryos.

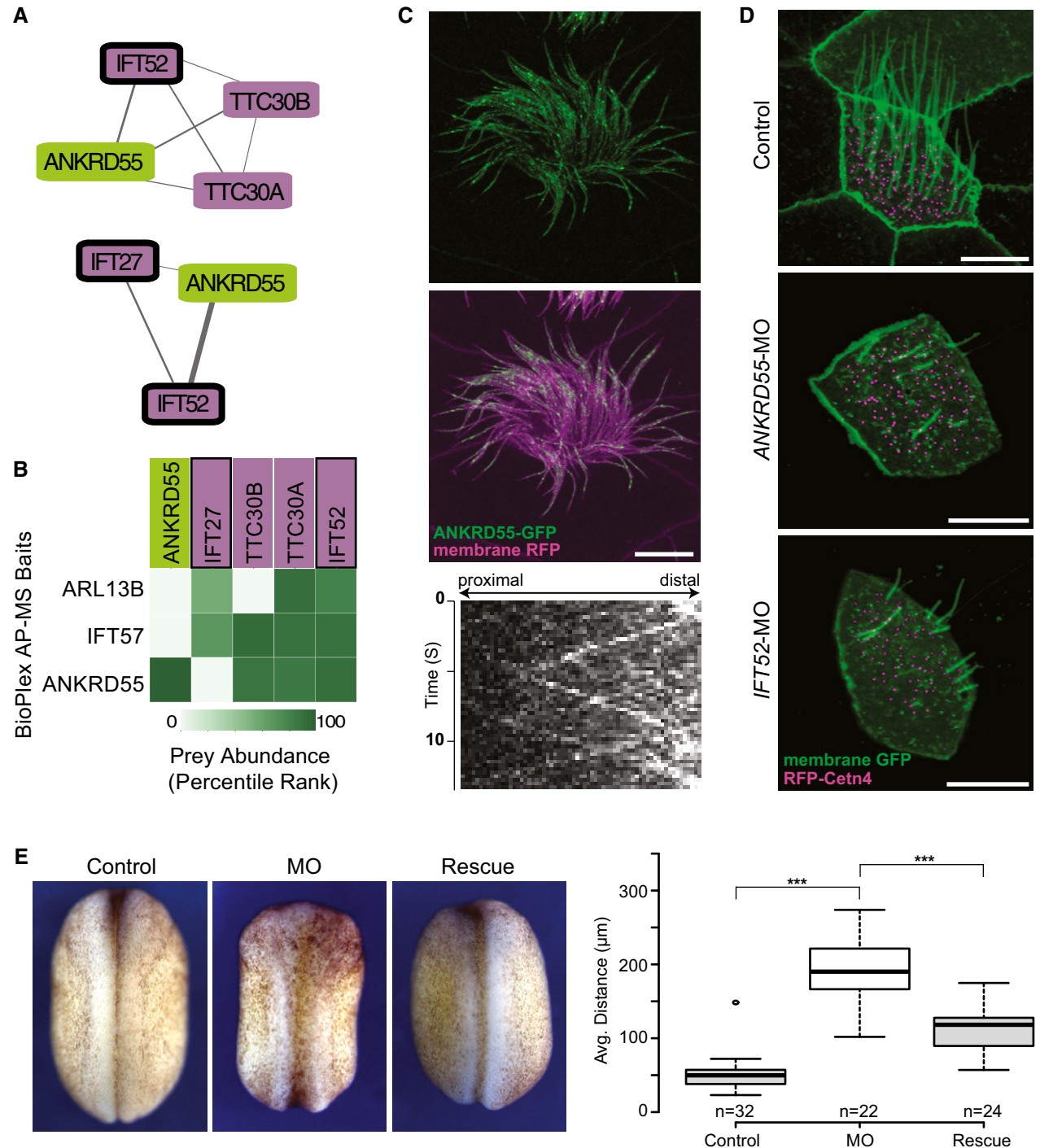

**Figure 7.  ANKRD55 is a new component of the intraflagellar transport (IFT) particle, is important for ciliogenesis, and has a role in neural tube closure.**

A   Network view of two IFT sub-complexes associated with ANKRD55.
B   Matrix of AP-MS experiments shows strong support for ANKRD55 association with known IFT proteins.
C   ANKRD55 localizes to cilia as predicted from co-complex interactions, as assayed *in vivo* in multi-ciliated *Xenopus laevis* epithelial cells. Scale bar: 10 μm. Each image is representative of 18 cells from six different embryos. Kymograph of ANKRD55 localized to cilia *in vivo* reveals rapid trafficking along the length of the cilia (representative out of 36 multi-ciliated cells).
D   Morpholino knockdown of *ANKRD55* results in reduced count and length of cilia, in a manner similar to the control IFT52 knockdown, supporting a role in ciliogenesis for ANKRD55. Scale bar: 10 μm. Each image is representative of 18 cells from six different embryos.
E   Dorsal view of stage 19 *X. laevis* embryos displays that *ANKRD55* knockdown causes neural tube closure defects that are rescued by wild-type *ANKRD55* mRNA. The Tukey box plot displays average distance between neural folds in control (*n* = 32), morphant (*n* = 22), and rescue (*n* = 24) embryos. ***$P < 0.0001$, two-sample Kolmogorov–Smirnov test. Boxes indicate median (inner band), first quartile (bottom) and third (top) quartile. Whiskers indicate 1.5 interquartile range. Dots indicate outliers.

# Discussion

Gaining a more complete understanding of the relationship between human genotypes and phenotypes will require improved maps of protein complexes as well as some understanding of their dynamic nature across cell types and across the spectrum from healthy to diseased tissue. Recent advances in proteomics now allow for the comparison of biological networks across different conditions to identify the dynamics of protein complex function (Ideker & Krogan, 2012; Kristensen *et al*, 2012). However, our ability to interpret these experiments is hindered by the lack of a complete picture of protein complexes. Here, we report a map that captures a significant portion of the core protein machinery in human cells. This map provides not only a framework on which to organize future experiments, but also provides immediate insight into broad classes of human diseases, including ciliopathies.

To produce this map, we described the re-analysis and integration of three large-scale protein interaction datasets. Our map advances the state of the art in three fundamental ways. First, we uncover additional signal in published AP-MS datasets using our weighted matrix model approach. We showed that the limited overlap of the input published networks is due in part to the computational analyses of the underlying experiments, which suggests more sophisticated analysis techniques may further uncover novel protein interactions. Integration across the datasets greatly enhanced the precision and recall of the final interaction network, in part by scoring prey–prey interactions, leading us to identify thousands of interactions which were previously unreported in the original publications, as for the synaptic bouton complex. This weighted matrix model approach should be of increasing importance because of its ability to elegantly compensate for and capitalize on off-target identifications in AP-MS datasets. The model's ability to take into account the frequency at which proteins are identified across experiments allows for the filtering out of non-specific and contaminating proteins found across datasets. It is likely the weighted matrix model approach will only become more powerful as additional datasets are available and can be combined to identify subtle trends across many experiments.

Second, we developed a machine learning framework that can easily incorporate new data types to build more comprehensive protein complex maps by integrating evidence across many experiments. In this work, we demonstrate the effectiveness of integrating multiple lines of evidence to identify protein interactions. Indeed, there is tremendous effort in the community to generate ever larger-scale maps of human protein interactions, and extensions to ongoing high-throughput interactome studies can be naturally incorporated into our protein complex discovery framework. We envision a continual expansion and refinement of this set of human protein complexes using the described pipeline as new high-throughput protein interaction experiments are published.

Third, we employed a clustering strategy that optimizes predicted complexes based on training complexes and allows efficient removal of false positives. To accomplish this, we developed a novel method for comparing the reconstructed protein complexes to a gold standard set of protein complexes, a problem that has proven difficult for the field. The solution we propose is formulated in a precision–recall framework based on cliques derived from the predicted clusters and gold standard set. This approach differs from previous solutions in that it generates a global comparison between clusters and the gold standard, rather than identifying the best match for each single cluster at a time. The comparison method is applicable whenever one wishes to compare two sets of sets, as it is general in nature and should be useful beyond comparing protein complexes. The overall clustering strategy allowed us to identify complexes that were otherwise missed by traditional clustering techniques.

The success of long-standing efforts to understand the genetic basis of human disease relies heavily on understanding the physical interactions of proteins. We demonstrate the value of our complex map for understanding human disease by featuring ciliopathy-related complexes. Through this analysis, we highlighted uncharacterized proteins, which we experimentally validated to be cilia-associated, as predicted by the map. We also knocked down one of these proteins, ANKRD55, and showed a disruption in ciliogenesis, which strongly suggests a role in ciliopathies. These results establish the ability of an integrated human protein complex map to identify new candidate disease genes, with potentially broad applicability to many human diseases.

# Materials and Methods

### Gold standard training and test set complexes

For training and evaluating our protein complex discovery pipeline, we used literature-curated complexes from the CORUM core set (Ruepp *et al*, 2010). We first removed redundancy from the CORUM set by merging complexes that had large overlap (Jaccard coefficient > 0.6). The set of complexes were then randomly split into two sets, labeled test and training. Due to proteins participating in multiple complexes, the randomly split sets were not fully disjoint. We dealt with the overlap of these two sets differently at the pairwise interaction and complex level, as follows:

For the purposes of training and evaluating our SVM classifier, we generated positive and negative pairwise protein interactions for both test and training sets. A positive protein interaction is defined as a pair of proteins that are part of the same complex. A negative protein interaction is defined as a pair of proteins that are both in the set of complexes but not part of the same complex. We addressed overlap here between the test and training (positive/negative) protein interactions by removing interactions from the training protein interaction sets that were shared in the test protein interaction sets, such that the sets were fully disjoint. Additionally, 112 pairs in the positive test set were not co-complex prior to our merge step and were removed.

For the analyses of protein complexes, in order to ensure that the test and training sets of complexes were disjoint, we removed entire complexes from the training set which shared any edge with a complex in the test set. In comparing the size distributions (the number of subunits per complex) between the training and test sets, we noticed a skew of larger complexes in the test set likely a result of our conservative approach of removing complexes from the training set. In order to better balance the training and test complex set size distributions, we first randomly split the test set into two and combined one half with the training complexes. We again applied our redundancy removal procedure, removing complexes from the training set which shared any edge with a complex in the

                                                                                                                        

test set. Similar to what has been done previously (Havugimana *et al*, 2012; Wan *et al*, 2015), we also removed complexes larger than 30 subunits from the test set so as not to skew performance measurements.

The final pairwise protein interaction training/test sets consisted of 27,665/15,575 and 2,543,855/2,867,914 positive and negative interactions, respectively. The final protein complex training/test sets consisted of 406/264 complexes. The complete lists of training/test interactions and complexes are available at the supporting web site (http://proteincomplexes.org).

### Calculating protein interaction features from the mass spectrometry datasets

We collected raw published features from three datasets, Wan *et al* (Wan *et al*, 2015), BioPlex (Huttlin *et al*, 2015), and Hein *et al* (Hein *et al*, 2015). All features from mass spectrometry experiments were calculated from raw data, and no feature was trained using any gold standard pairs prior to input into our machine learning framework. Wan fractionation features included four measures of co-fractionation from 5,162 MS experiments as well as 19 lines of evidence from HumanNet (Lee *et al*, 2011) and two additional AP-MS datasets (Guruharsha *et al*, 2011; Malovannaya *et al*, 2011). Specifically, the co-fractionation measures, as described previously (Havugimana *et al*, 2012; Wan *et al*, 2015), included a Poisson noise Pearson correlation coefficient, a weighted cross-correlation, a co-apex score, and a MS1 ion intensity distance metric. Each co-fractionation measure was applied to each fractionation experiment, totaling 220 features. As described in Wan *et al*, pairs of proteins were filtered to ensure co-fractionation measures were > 0.5 in at least two species.

Additional features were taken from Wan *et al* (2015). In summary, HumanNet features were originally downloaded from http://www.functionalnet.org/humannet/download.html (file: HumanNet.v1.join.txt). We excluded HS-LC (human literature-curated) and HS-CC (human co-citation) evidence codes to remove circularity in the training process. HumanNet features were only included for pairs of proteins that had substantial co-fractionation evidence, specifically co-fractionation measure greater than 0.5 in at least two species. This resulted in 35,028 pairs with HumanNet evidence (1.3% of the total number of pairs in the feature matrix). The additional AP-MS fly feature, HGSCore value, was downloaded from supplemental table S3 in Guruharsha *et al* (2011). The additional AP-MS human feature was based on the MEMOs (core modules) certainty assignments "approved", "provisional", and "temporary" downloaded from supplemental file S1 in Malovannaya *et al* (2011), assigning the scores 10, 3, and 1, respectively. As was done with the HumanNet features, features from Guruharsha *et al* and Malovannaya *et al* were only included for pairs of proteins which had co-fractionation evidence available.

BioPlex AP-MS features were downloaded from:http://wren.hms.harvard.edu/bioplex/data/cdf/150408_CDF_STAR_GRAPH_Ver2594.cdf

Specifically, we used the following nine features: NWD Score, Z Score, Plate Z Score, Entropy, Unique Peptide Bins, Ratio, Total PSMs, Ratio Total PSM's, and Unique: Total Peptide Ratio. For the Hein AP-MS data, the features prey.bait.correlation, valid.values, log10.prey.bait.ratio, and log10.prey.bait.expression.ratio were taken from supplemental table S2 in Hein *et al* (2015). In the case of

multiple entries for a given protein pair, the mean value was used across the experiments.

We generated two additional features for both the BioPlex and Hein AP-MS datasets based on a weighted matrix model interpretation, specifically, the number of experiments a pair of proteins is observed together (pair_count) as well as a $-1*\log(P\text{-value})$ of two proteins being observed together at random across all AP-MS experiments, as calculated using the hypergeometric distribution as previously described (Hart *et al*, 2007).

Missing values for any of the features were set to 0.0 in the final feature matrix.

### Accurate learning of pairwise protein interactions

Given this feature matrix, we next proceeded to train a SVM protein interaction classifier. We scaled the feature values using LIBSVM's (Chang & Lin, 2011) `svm-scale` to avoid features with larger numeric range from dominating the classifier. We performed a parameter sweep of the SVM C and gamma parameters using LIBSVM's cross-validation `grid.py` utility. Training and prediction were calculated using LIBSVM's `svm-train` and `svm-predict` tools with the "probability estimates" option set to true. Finally, we applied the SVM classifier to all pairs of proteins for which we had data, thereby generating a protein interaction network in which edge weights between protein nodes were set to the SVM's probability estimate for interacting. We repeated this procedure for combinations of features including only features for individual publications, as well as combinations found in Figs 2A and EV2A–C. To calculate precision–recall curves, we used the python scikit-learn machine learning package (Buitinck *et al*, 2013).

### Identifying protein complexes by clustering the interaction network

We applied a two-stage clustering approach to the protein interaction network to identify clusters of densely interacting proteins, representing our best estimates of protein complexes. First, we sorted the edges of the protein interaction network by their interaction probabilities and selected the top *f* percent of edges, where *f* is a parameter in the range of [0.008, 0.01, 0.015, 0.02, 0.025, 0.03, 0.05] determined by a parameter sweep described below. We applied the ClusterOne algorithm (Nepusz *et al*, 2012) to the resulting interaction network, specifying minimum size parameter = 2, seed method parameter = "nodes", density in the range [0.2, 0.25, 0.3, 0.35, 0.4], and overlap in the range [0.6, 0.7, 0.8]. For each cluster produced by the ClusterOne algorithm, we refined the clustering by performing a second round of clustering using the MCL algorithm (Enright *et al*, 2002), specifying the MCL parameter inflation (−I) to be in the range [1.2, 2, 3, 4, 5, 7, 9, 11, 15]. In parallel, we refined each ClusterOne cluster using an alternate second-stage clustering algorithm, the Newman method (Newman, 2004). Finally, we removed any protein from the resulting clusters that did not have an edge weight to the remaining proteins in the cluster scoring above the filter parameter, *f*, which occasionally, although rarely, arose through the action of the MCL algorithm.

To objectively optimize the choice of clustering parameters, we performed the two-stage clustering process for each combination of parameters, varying *f*, density, overlap, and inflation and selected

the cluster set that maximized the *F*-Grand *k*-clique measure compared to the training set of literature-curated complexes. The best-scoring parameters for ClusterOne + MCL were size: 2, density: 0.2, overlap: 0.7, seed_method: nodes, inflation: 7, and *f*: 0.03. The final parameters for ClusterOne + Newman were size: 2, density: 0.4, overlap: 0.7, seed_method: nodes, and *f*: 0.02. Edges that passed the *f* filter corresponding to an interaction probability of 0.26509 were considered high confidence. Finally, we combined the best-scoring two-stage clustering sets (i.e., the union of the best performing ClusterOne + MCL and ClusterOne + Newman sets) to form the final estimate of protein complexes.

## Measuring accuracy of the protein complex map by the *k*-clique method

Guiding and assessing the accuracy of the reconstructed complexes requires comparison with a gold standard set of known complexes. However, comparing sets of complexes to known complexes (or more generally, comparing sets of sets with each other), is ill defined due to the problem of first deciding which sets should be compared and second, to the incomparable nature of specific matches. For instance, given two non-overlapping complexes, one of size 3 and one of size 20, it is difficult to assess whether an exact match of the complex of size 3 should be given more weight than a partial match of the complex of size 20 (e.g., with 17 out of 20 correct). Many complex–complex comparison metrics (Bader & Hogue, 2003; Brohée & van Helden, 2006; Song & Singh, 2009) have attempted to address this issue, but they are often difficult to interpret and may lead to false minima in the parameter landscape, in part because they require a mapping procedure to determine which specific gold standard complexes match up with specific reconstructed complexes.

In order to more systematically address these issues, we invented a new class of similarity metrics, *k*-cliques, for comparing sets of complexes in a formal precision–recall framework. Specifically, our approach is based on the matching of cliques within the set of all possible cliques between predicted complexes and benchmark complexes. Cliques range from size 2 (pairwise protein interactions) through *n*, where *n* is the size of the largest predicted complex. The approach allows for precision and recall values to be calculated unambiguously (because a clique is either present or absent in a given set) for each clique size, *k*, and averaged to determine a single performance metric (here, the *F*-Grand metric, corresponding to the average across all clique sizes of the harmonic mean of precision and recall). An important feature of the *k*-clique approach is that it focuses the evaluation on protein interactions, rather than the proteins themselves, providing a unique perspective on set comparisons. In addition, while other approaches suffer from evaluating each complex individually and often require a cluster reduction step in which similar clusters are combined to avoid potential skew, for example, as caused by prediction of sub-complexes of larger complexes, the *k*-clique approach compares complexes on a global level and naturally deals with potential skew by only evaluating on the unique set of cliques for all predicted complexes. Finally, there is no need to determine a unique mapping between each predicted and benchmark complex, thus avoiding mapping-induced ambiguity.

In detail, let *C* be a set of predicted complexes $\{c_1, c_2, \ldots, c_n\}$ and *D* be a set of gold standard complexes $\{d_1, d_2, \ldots, d_m\}$, where $c_i$ and $d_j$ are an individual predicted complex and gold standard complex, respectively. Let $Q_D$ be the set of protein identifiers in *D* (equation 1).

$$Q_D = \bigcup_{|D|} d_j \tag{1}$$

*P* represents the powerset (set of all subsets) and $P_k$ represents the powerset of a given size (e.g., $k = 2$, all pairwise combinations; $k = 3$, all triplet combinations; *etc.*). $A_k$ (equation 2) represents the set of all size *k* cliques in the predicted clusters, *C*. An additional condition on $A_k$ is that the individual cliques overlap with proteins in the gold standard set ($Q_D$, equation 1), so we only evaluate on proteins that have known complex memberships. The rationale for this is so we do not penalize novel predicted complexes as false positives. Similarly, $B_k$ (equation 3) represents the set of all size *k* cliques in the gold standard complexes set *D*. Note, there is no condition on $B_k$ in terms of protein membership as was done with $A_k$. This results in an absolute recall measure and evaluates on all complexes in the gold standard regardless of whether or not there is ample data for those proteins.

$$A_k = \bigcup_{c_i \in C} (P_k(c_i) \cap P(Q_D)) \tag{2}$$

$$B_k = \bigcup_{d_j \in D} (P_k(d_j)) \tag{3}$$

Definitions of $A_k$ and $B_k$ now provide us with a way to compare size *k* cliques in predicted clusters to size *k* cliques in gold standard complexes in a precision–recall framework. Equations (4–6) describe the operations of determining true positives ($TP_k$), false positives ($FP_k$), and false negatives ($FN_k$), respectively, for a given clique size *k*.

$$TP_k = |A_k \cap B_k| \tag{4}$$

$$FP_k = |A_k \backslash B_k| \tag{5}$$

$$FN_k = |B_k \backslash A_k| \tag{6}$$

Equations (7) and (8) define precision ($P_k$) and recall ($R_k$), and equation (9) defines *F*-measure ($F_k$) as the harmonic mean of $P_k$ and $R_k$.

$$P_k = \frac{TP_k}{TP_k + FP_k} \tag{7}$$

$$R_k = \frac{TP_k}{TP_k + FN_k} \tag{8}$$

$$F_k = 2 \times \frac{P_k \times R_k}{P_k + R_k} \tag{9}$$

Finally, we define a global *F*-measure (*F-Grand*, equation 10) as the mean of $F_k$'s, iterating over clique sizes of *k* from 2 to *K* where *K* is the max cluster size of the predicted clustering set *C*.

$$F_{\text{grand}} = \frac{\sum_{k=2}^{K} F_k}{K - 1} \tag{10}$$

Additionally, we define an alternative global measure that defines weights for each $P_k$ and $R_k$ by the number of clusters, $w_k$, with size $\geq k$. This allows for the mitigation of potential bias created by large clique sizes only having a few contributing clusters.

$$R_{\text{weighted}} = \frac{\sum_{k=2}^{K} w_k * R_k}{\sum_{k=2}^{K} w_k} \tag{11}$$

$$P_{\text{weighted}} = \frac{\sum_{k=2}^{K} w_k * P_k}{\sum_{k=2}^{K} w_k} \tag{12}$$

$$F_{\text{weighted}} = 2 \times \frac{P_{\text{weighted}} \times R_{\text{weighted}}}{P_{\text{weighted}} + R_{\text{weighted}}} \tag{13}$$

In practice, the sizes of the clique sets (equations 2 and 3) are quite large and computationally intractable to calculate. We therefore randomly sample 10,000 cliques from $A_k$ and $B_k$ when evaluating true-positive, false-positive, and false-negative values (equations 4–6). Additionally, we add a pseudo-count of 0.00001 to true-positive, false-positive, and false-negative values when calculating precision and recall (equations 7 and 8). We have implemented a script to calculate the *F*-weighted *k*-clique score that is available in our project GitHub repository. An example command line is as follows:

```
python complex_comparison.py --cluster_predic-
tions hu_MAP.txt --gold_standard testComplexes.txt
```

**Measuring overlap with independent protein interaction datasets**

In order to assess agreement between our complex map and other protein interaction datasets, we compared the observed overlap of protein interactions to the overlap expected by chance. We chose three datasets that were not integrated into our protein complex discovery pipeline, the yeast two-hybrid dataset from Rolland *et al* (2014), released but unpublished interactions (06/12/2015) from the BioPlex project (BioPlex), and an inter-protein cross-linking (1% FDR) dataset from Liu *et al* (2015).

For the unpublished BioPlex dataset, we removed all interactions that overlapped with the original published BioPlex interaction set to ensure a disjoint set with our training dataset. For the Liu *et al* cross-linking dataset, we considered a non-redundant subset by collapsing all inter-protein cross-link interactions for each pair of proteins to just a single interaction.

For each interaction dataset, we generated 1,000 random interaction sets by randomly selecting *M* pairs of proteins where *M* is the number of interactions in that dataset. We then compared the overlap of interactions from our complex map with the random interaction sets to determine a random distribution and calculated a z-score for the overlap of our complex map and the original interaction dataset relative to the random distribution.

**Synaptic bouton complex expression analysis**

We downloaded human normalized microarray datasets H0351.2001, H0351.2002, H0351.1009, H0351.1012, H0351.1015,

H0351.1016 from the Allen Brain Map (Hawrylycz *et al*, 2012) [downloaded from: http://human.brain-map.org/static/download]. For each gene in the synaptic bouton complex, we averaged expression values across corresponding probes and calculated Pearson correlation coefficients for each pair of genes. For comparison to a random background distribution, we randomly selected 60 probes from the microarray datasets and calculated Pearson correlation coefficients between the random probes and the genes in the synaptic bouton complex.

For tissue expression analysis of synaptic bouton complex genes, we used RNA-sequencing data for 32 tissues from the Human Protein Atlas (Uhlén *et al*, 2015) [downloaded: http://www.prote inatlas.org/download/rna_tissue.csv.zip].

**Calculations of tissue specificity, annotation enrichment, and coverage**

For comparing tissue specificity, we used reported RNA tissue category assignments from the Human Protein Atlas (Uhlén *et al*, 2015) [downloaded: http://www.proteinatlas.org/download/prote inatlas.tab.gz]. We mapped proteins to HPA entries with RNA tissue category classifications, considering either the entire human proteome, the union of proteins from protein interaction networks of Wan *et al*, BioPlex, and Hein *et al*, or the proteins in our final complex map. In order to calculate enriched annotations for each complex, we applied g:Profiler (Reimand *et al*, 2016) with a FDR (Benjamini–Hochberg) *P*-value correction per each complex and excluded electronic annotations from consideration. We used the complete set of proteins in the final protein interaction network as the statistical background. Additionally, we produced a random set of complexes by permuting the complex membership of proteins in hu.MAP and calculating enriched annotations for each shuffled complex as described above. We then used this set of significantly enriched annotations on shuffled complexes to calculate a false discovery rate of 5% on hu.MAP complex enrichments. In order to calculate coverage of diseases, we mapped OMIM annotations (Amberger *et al*, 2015) onto Disease Ontology (Schriml *et al*, 2012) terms and then selected the top eight disease categories as well as the term "ciliopathies". We then mapped proteins from our complex map, the Wan *et al* complex map [downloaded: supplementary table S2 from Wan *et al* (2015)], the BioPlex complex map [downloaded: supplemental table S3 from Huttlin *et al* (2015)], and Hein *et al* protein interaction network [downloaded: supplemental table from Hein *et al* (2015)] onto the Disease Ontology terms. We also evaluated coverage of proteins in the SysCilia Gold Standard Version 1 set of cilia-related proteins (downloaded: http://www.syscilia.org/goldstandard.shtml) (van Dam *et al*, 2013).

**Calculation of prey abundance and network visualization**

To calculate percentile ranks of prey abundance for AP-MS raw data, we used the prey abundance measures "zscore" and "prey.bait.correlation" from BioPlex and Hein *et al*, respectively. We ordered each set and calculated the rank percentile using SciPy stats.percentileofscore (Jones *et al*, 2015) for each pair in the list. Networks of protein complexes were visualized using Cytoscape 3.2.1 (Shannon *et al*, 2003).

## Morpholinos and mRNA synthesis

Morpholino antisense oligonucleotides (MOs) were purchased from Gene Tools. The *ANKRD55* MO was designed to block splicing using the sequence 5′-TCTGAATCACCTTGAAGCACAAAGA-3′. We used previously validated MOs for *JBTS17*, 5′-TCTTCTTGATCCACTTA CTTTTCCC-3′ (Toriyama *et al*, 2016); and *IFT52*, 5′-AAGCAATC TGTTTGTTGACTCCCAT-3′ (Dammermann *et al*, 2009). Full-length *ANKRD55* cDNA (identified from Xenbase, www.xenbase.org) was amplified from a *Xenopus* cDNA library and subcloned into the vector pCS10R (derived from pCS107 expression vector) fused with C-terminal GFP. The human *CLUAP1* open reading frame was obtained from the Human ORFeome collection V7.1 and subcloned into the pCS10R-mCherry vector. Capped mRNAs were synthesized using mMESSAGE mMACHINE (Ambion). mRNAs and MOs were injected into two ventral blastomeres or two dorsal blastomeres at the 4-cell stage to target the epidermis or the neural tissues, respectively. We used each mRNA or MO at the following dosages: *ANKRD55* MO (30 ng for the epidermis and 20 ng for the neural plate), *JBTS17* MO (20 ng), *IFT52* MO (40 ng), *ANKRD55*-GFP mRNA (75 pg), *ANKRD55* mRNA (350 pg for neural tube closure rescue experiment), membrane RFP mRNA (50 pg), and mCherry-*CLUAP1* (100 pg).

## Imaging and analysis

For high-speed live imaging, *Xenopus* embryos injected with *ANKRD55*-GFP and mCherry-*CLUAP1* mRNA were anaesthetized with 0.005% benzocaine at stage 26. High-speed *in vivo* imaging was acquired on a Nikon Eclipse Ti confocal microscope with a 63×/1.4 oil immersion objective at 0.267 s per frame. Kymographs were calculated using Fiji (Schindelin *et al*, 2012). Confocal images were collected with an LSM700 inverted confocal microscope (Carl Zeiss) with a Plan-APOCHROMAT 63×/1.4 oil immersion objective. Bright field images were collected using a Zeiss Axio Zoom V16 stereo microscope with Carl Zeiss Axiocam HRc color microscope camera. Neural tube closure quantification was performed using Fiji. A two-sample Kolmogorov–Smirnov test was used to compare distributions of control, morphant, and rescue embryos. Sample sizes were selected sufficient to determine moderate effects. Embryos were randomly selected from multiple clutches, and cells were randomly selected from individual embryos. No blinding to treatment was employed.

## Plasmids

*CCDC138*, *CCDC61*, *TBC1D31*, *FOPNL*, *MIB1*, *PIBF1*, and *SSX2IP* entry ORF clones were obtained from the DNASU Plasmid Repository (Seiler *et al*, 2014; Grant *et al*, 2015). *Xenopus laevis* cDNA was prepared by reverse transcription (SuperScriptIII First-Strand Synthesis, Invitrogen), and *KIAA1328*, *WDR90*, *TTLL5* cDNAs were PCR-amplified from the library using the following primers:
*WDR90F*: caccATGGCTGGAGTCTGGCAG
*WDR90R*: TGAATTCTGAATGTCCCACAC
*TTLL5F*: caccATGCCCGAAATGTTGCC
*TTLL5R*: TTTTCTTTGCCCTTTACTGTCGA
*KIAA1328F*: caccATGGATTTACAGAGGCAGCAAG
*KIAA1328R*: ACAAATGAAGAAGATCTCCTCTAACATC

PCR products were subcloned into Gateway ENTRY clones (pENTR/D-TOPO Cloning Kit, Life Technologies). Destination vectors were modified from destination vector Pcsegfpdest (a gift from the Lawson laboratory) by inserting the α-tubulin promoter between the SalI and BamHI sites. Fluorescence protein-tagged expression plasmids were constructed using the LR reaction on entry clones and destination vectors with the Gateway LR Clonase II Enzyme mix (Life Technologies). Expression plasmids (40 pg) with centrin-BFP mRNA (100 pg) were co-injected into the ventral blastomeres of *Xenopus* embryos at the 4-cell stage and imaged at stage 27. Sample sizes were chosen to obtain representative images.

## *Xenopus* embryos

*Xenopus* embryo manipulations and injections were carried out using standard protocols. All experiments were performed following animal ethics guidelines of the University of Texas at Austin, protocol number AUP-2015-00160.

## Data availability

Protein interactions and complexes have been deposited in the BioGRID database (Stark 2006) (https://thebiogrid.org/dataset/ma rcotte2017) and can be searched interactively or downloaded at http://proteincomplexes.org. Supporting computer code for the full protein interaction mapping and analysis pipeline is available at: https://github.com/marcottelab/protein_complex_maps_public.

**Expanded View** for this article is available online.

## Acknowledgements
This work was supported by grants from the NIH (F32 GM112495 to K.D.; 1R01 HL117164 to J.B.W.; R21 GM119021, R01 HD085901 to J.B.W. and E.M.M.; and DP1 GM106408, R01 DK110520, R35 GM122480 to E.M.M.), NSF and CPRIT (to E.M.M.), and the Welch foundation (F-1515, to E.M.M.).

## Author contributions
KD and EMM designed project. KD developed code and performed data analysis. CL, RLH, FT, YM, and JBW designed and performed validation experiments. BB contributed code. CDM performed analysis. KD, JBW, and EMM drafted manuscript. All authors discussed results and contributed edits.

## Conflict of interest
The authors declare that they have no conflict of interest.

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
