## [Review Process File · Molecular Systems Biology]

Integration of over 9,000 mass spectrometry experiments builds a global map of human protein complexes

Kevin Drew, Chanjae Lee, Mr. Ryan Huizar, Mr. Fan Tu, Blake Borgeson, Ms. Claire McWhite, Yun Ma, John Wallingford and Edward Marcotte

Corresponding author: Edward Marcotte, University of Texas at Austin

Review timeline:	Submission date:	07 December 2016
	Editorial Decision:	23 January 2017
	Revision received:	21 March 2017
	Editorial Decision:	18 April 2017
	Revision received:	02 May 2017
	Accepted:	12 May 2017

Editor: Maria Polychronidou

Transaction Report:

1st Editorial Decision

23 January 2017

Thank you again for submitting your work to Molecular Systems Biology. We have now heard back from the three referees who agreed to evaluate your study. As you will see below, the reviewers appreciate that an improved characterization of human protein interactions is timely and useful for the field. However, they list several issues, which we would ask you to address in a major revision.

Without repeating all the points listed below, one of the more fundamental issues refers to the potential inclusion of erroneous interactions by the matrix expansion (reviewer #2, point #1). Moreover, the reviewers raise several technical issues, which should be carefully addressed.

During our pre-decision cross-commenting process (in which the referees are given the chance to comment on each other's reports), reviewer #3 mentioned that re-analyzing all MS datasets as reviewer #1 suggests (point III), requires a rather substantial effort that in his/her experience it is not expected to considerably change the final outcome. As such, we think that this additional analysis is not mandatory. Of course, we would not be opposed to the inclusion of such analyses i.e. if you have already performed them or feel inclined to do so.

REFeree REPORTS

Reviewer #1:

This manuscript describes a bioinformatics pipeline that incorporates a number of novel features to decipher large scale protein-protein interaction (PPI). Namely, the authors performed a matrix

model to AP-MS experiments (instead of the typically adopted spoke model), SVM classifier and two-stage clustering. They had also combined PPI datasets from different sources. As a result, they were able to uncover protein complexes that are missed in the conventional pipeline, in addition to discovering hidden components and complexes by further combining with other correlation analysis such as co-expression and co-appearance in diseases, based on the principles of guilt-by-dissociation.

PPI is a very important facet of proteomics because protein networks lie at the core of cellular functions. Protein complexes are dynamic entities that can be rewired, responding to external stimuli, development, time and space. Therefore, the study is very important and of potential interest. Moreover, the authors have attempted to address a crucial issue in the field of interaction proteomics i.e. the limited overlap and non-comprehensiveness of PPI data. Though the proteomics field has an abundance of interaction proteomics studies, bioinformatics pipelines addressing limitations in this field is rather limited.

The following are my comments:

(I.) By using a matrix model to AP-MS data, prey-prey interactions that would have been missed by only using spoke model alone are detected. Thus, it increases the overlap of PPI between the 3 large datasets used. However, the authors had decided to choose N = number of interactions reported in the original studies. That means the authors would have to discard some bait-prey pairs from the original studies. Why is that so?

(II.) Yeast-2-hybrid studies are prevalent in PPI studies. Would be useful to incorporate Y2H studies in this pipeline?

(III.) In this pipeline, original features derived from different datasets (labs) are used. However, different labs performed MS raw data analysis, as well as the subsequent statistical analysis in different ways. This is especially true for proteomics data where the MS raw data need to go through several stages of analyses and may result in discrepancy in results. Would the results be more accurate and precise if all three MS datasets were re-analyzed from scratch? This should be possible as all proteomics data should be in public depositories.

(IV.) Is there an explanation that the 4 baits co-precipitate all 60 subunits of synaptic bouton complex but they do not pull down each other?

Reviewer #2:

The manuscript by Drew et al. attempts to exploits large recent datasets aiming at identifying protein-protein interactions by affinity purification coupled with mass spectrometry (AP-MS) or co-fractionation coupled to mass spectrometry (CF-MS) in order to better define protein complexes.

A better characterization of protein complexes is very much needed, as previous resources (MIPS, CORUM) have not kept pace with high-throughput approaches. The current manuscript attempts to bridge this gap.

The authors reanalyze 3 large datasets (Huttlin et al., Hein et al., Wan et al., all published in 2015), that together encompass "9000 mass spectrometry experiments". They claim higher coverage resulting from this analysis, which in part is due to the re-scoring of the interactions in both the Huttlin and the Hein data, moving away from the "spoke" model used in the initial analysis by the authors. Here, the authors apply a matrix model, scoring prey-prey interactions through a hypergeometric distribution model. Using the top N "interactions" scored in each of the AP-MS datasets with this model as compared to the spoke model used by the initial publications led to a bigger overlap between the Hein and Huttlin datasets.

The authors next combined this rescored AP-MS set with the Wan et al. dataset (to which it should be orthogonal), and used machine learning approaches and clustering of pairwise interactions to identify protein complexes. Comparison to curated complexes in CORUM enabled benchmarking of

the different scoring / clustering approaches.

The manuscript itself is well-written, but I have major reservations with the complexes obtained by the approach (from browsing the author's reference site), which are described below.

1. While the matrix expansion certainly will increase the coverage of the datasets, it also introduces issues which are apparently not considered here by the authors. This is particularly apparent when looking at protein complexes for which A binds to either B or B', but never to both. In many instances (on the authors' database), there will be a "complex" that contains A-B-B', which is not biochemically realistic. This is particularly obvious with paralogous proteins (so in theory these could be flagged computationally), but it also happens when B and B' are only related by the interaction motif for A. A spoke expansion model would of course require at least B or B' (and preferably both) to be able to discern the mutually exclusive nature of the interactions, but at least it would have a chance to get at the correct answer. After browsing the website, I am worried that many of the new interactions / new complexes are erroneously called.

2. The current version of the map should certainly not be called "definitive" (as is currently subtitled on the website), especially as it only integrates data from 3 studies, leaving aside a huge fraction of the protein interaction literature (let alone other issues with the dataset). In particular, several high quality AP-MS studies of more moderate scale could have helped expand the coverage and accuracy of the maps (and perhaps escape issues with matrix expansion). In the discussion, the authors refer to their manuscript as "a framework", which is certainly more appropriate, notwithstanding the issues with the current version of the manuscript.

3. This is certainly not the authors' fault, but the "gold standard" that is CORUM does not have "complexes" which are all similarly supported by experimental evidence. Some of the "complexes" are based on a single publication/experiment type (including older AP-MS experiments which may or may not have been performed with appropriate controls). Annotation is also in some cases problematic, and in some cases, the same type of A+B or A+B' situation (even when acknowledged in the original manuscript) is annotated as an A, B, B' complex even in CORUM. This type of situation would, of course, influence the benchmarking of the author's dataset. It is not clear what the step of merging complexes based on the Jaccard coefficient of shared components does to the overall reliability of the CORUM data.

4. It is not clear to me how the Hein data, which has triplicates analysis of each bait, was analyzed in relation to the Huttlin data which only analyzed a single bait (did repeated observations influence the weight?).

5. On the website, I realize that the authors are listing the evidence source for the complexes. However, it is unclear what the "hein" and "bioplex" alone sources indicate in the absence of the definition of the bait-prey pairs that have lead to the conclusion that there is indeed a complex being formed.

6. The identification of "complexes" by their observations only as "preys" is certainly intriguing. This being said, the authors would need to demonstrate experimentally that somehow proteins identified solely on the basis of their recovery as preys are indeed forming a complex. Many of the proteins in this "synaptic bouton complex" in Figure 3 are associating with different endosomal populations (e.g. all the Rab proteins, and syntaxins) or microtubules (e.g. Map proteins) and while it is not impossible that they indeed form a complex, another possibility is that they simply co-precipitate with sets of baits that are localized to the trafficking machinery and/or the cytoskeleton (can the authors exclude that a membrane-bound organelle/vesicle is co-precipitated?). This needs to be experimentally validated.

7. I was not clear from the description of Figures 4-6 whether an attempt was made at directly comparing the authors' network with those of Boldt and Gupta who performed large scale directed proteomics studies to profile the centrosome and cilium (e.g. CCDC138 was uncovered - though not validated - in the Gupta paper).

8. On Figures 5 to 7, it would have been nice to show the evidence source on the edges to better highlight the importance of the new dataset in comparison to the individual datasets in generating new knowledge. For example, it seems (looking at the individual datasets) that all the raw data about WDR90 comes from Hein et al. (the same seems to be true for KIAA1328 while the converse - all hits from Huttlin - exist for ANKRD55). The authors should make a better case for the specific need for Hu.MAP to make these hypotheses.

Minor comments

1- It would be nice to present legends for the tables (as well as more explicit titles)

- 2- Pictures of ciliopathies from Figure 5 are not relevant as presented; remove, unless there is some particular point that needs to be made (and which is currently not explained well).
- 3- Since the same type of benchmarking was also used in the selection of the scoring for Wan et al. (as far as I can tell), the authors should probably expand on how they avoid circularity in the scoring for those interactions which are supported by the CF-MS data.

Reviewer #3:

This manuscript describes an effort to assemble a catalogue of human protein complexes by integrating data from different sources, including a large number of mass spectrometry based measurements that are indicative of protein protein interaction. Drew and colleagues have compiled features from 3 different recent studies and integrated these using an SVM based approach. The ranked list of protein interactions were then clustered into protein complexes. The pairwise interactions and derived protein complexes were properly benchmarked against unseen data. This updated view of human interactions and complexes was then used to study in more detail ciliary proteins, identifying novel components of ciliary protein complexes that were validated by localization. A novel member of the IFT complex was also discovered and validated experimentally.

This work is most related with previous efforts from the same group at consolidating and integrating evidences for human protein interactions (e.g. Lee et al. Genome Research 2011). This is a very timely effort since so much new information has come out on human protein interactions and each individual study has been kept isolated. There is a great opportunity here to provide an updated human interaction/complex version that is of much higher coverage and accuracy than other previous attempts. This group has a lot of past experience in these approaches and the work is well done. However, I have some concerns about some of the technical aspects, in particular on not being sufficiently explicit about the use of some of the data sources such as the yeast, work and fly data or the human co-expression information. From a presentation point of view the article could also focus less of some of the computational details and perhaps more on the delivered interactions and complexes resource.

Major concern:

1- Superficially the authors appear to integrate 3 mass spectrometry based studies as stated throughout the manuscript. However, in the methods section one learns that the features used from the Wan et al. study include 19 features from HumanNet that include a very diverse set of data types and orthology transferred data. For example these include the large survey of genetic interactions from yeast, human co-expression information, human co-IP and human yeast-two-hybrid data, among several others. This is very misleading. I assume that the authors did not make this explicit in the manuscript in order to simplify the narrative but I think it is important to inform the readers more clearly about this. In particular, it may give an incorrect sense of relative value of the co-fractionation experiments to the readers.

1.1 - The authors should consider the co-fractionating features separately in their study. In figure 1E, this would mean clearly showing that some features are transferred by orthology and include additional data types such as gene expression, Y2H, genetic-interactions, etc. In Figure 2A it is again important to benchmark separately the features from co-fractionation from the HumanNet features from the co-IP data from the other 2 studies.

1.2 - From the HumanNet set of features, the authors should not consider any feature that is curated from the literature or based on co-citation even for species other than human. Stable protein complexes tend to be highly conserved across species so literature curated complexes from other species should be avoided in the training. This includes at least the features SC-LC, SC-CC, CE-LC and CE-CC.

1.3 - There are a large number of protein interactions in human derived from Y2H experiments. Together with the 3 works integrated in this study the Y2H networks from Vidal and colleagues would also be one very large set of interactions that has been published recently. Some of the Y2H interactions are included as part of the HumanNet features, which as stated above, I find misleading. The most recent large Y2H study was used by the authors to validate the predictions. Why haven't the authors consider them all as a feature? Was it again mostly to simplify the narrative of the manuscript or did they not provide additional predictive power?

1.4 - How up-to-date are the HumanNet features? Are these just taken directly as were done for Lee

et al. ?

Minor concerns

1 - How were the missing values handled for the SVM ? The feature matrix is certainly extremely sparse. Were all the missing values treated similarly ? In some cases the absence of interaction could be interpreted as a negative information but this will not be equally so for all methods and all interactions.

2 - The title is somewhat of a gimmick. The network integrates other experiments beyond MS data and the fact that there were 9000 MS "experiments" integrated is itself not the result. The authors are using genetic interaction data from yeast that include now around 539,710 interactions according to biogrid or gene expression data for human that probably also includes a very high number of samples. I would suggest to remove the reference to the mass spectrometry data from the title. Perhaps focusing on the comprehensiveness of the derived interaction network and core complexes.

3 - Some of the benchmarking sections could be described more succinctly in the text. In particular the evaluation of protein complexes using the k-cliques strategy. It is very important to properly benchmark the interactions and complexes and I think this has been well done in this work. However, for a wider audience this section could be reduced.

4 - Having a look at the list of predicted complexes it looks like many complexes have just 2 subunits. It could be useful for readers to have a figure showing the number of complexes having N number of subunits.

5 - In figure 5, do the authors have permission to use the phenotype figures from the other articles ?

1st Revision - authors' response

21 March 2017

We would first like to thank the reviewers for their thoughtful comments. We feel that their comments have helped us improve our manuscript specifically in areas that may have potentially caused confusion for readers.

Reviewer #1:

This manuscript describes a bioinformatics pipeline that incorporates a number of novel features to decipher large scale protein-protein interaction (PPI). Namely, the authors performed a matrix model to AP-MS experiments (instead of the typically adopted spoke model), SVM classifier and two-stage clustering. They had also combined PPI datasets from different sources. As a result, they were able to uncover protein complexes that are missed in the conventional pipeline, in addition to discovering hidden components and complexes by further combining with other correlation analysis such as co-expression and co-appearance in diseases, based on the principles of guilt-by-dissociation.

PPI is a very important facet of proteomics because protein networks lie at the core of cellular functions. Protein complexes are dynamic entities that can be rewired, responding to external stimuli, development, time and space. Therefore, the study is very important and of potential interest. Moreover, the authors have attempted to address a crucial issue in the field of interaction proteomics i.e. the limited overlap and non-comprehensiveness of PPI data. Though the proteomics field has an abundance of interaction proteomics studies, bioinformatics pipelines addressing limitations in this field is rather limited.

The following are my comments:

(I.) By using a matrix model to AP-MS data, prey-prey interactions that would have been missed by only using spoke model alone are detected. Thus, it increases the overlap of PPI between the 3 large datasets used. However, the authors had decided to choose N = number of interactions reported in the original studies. That means the authors would have to discard some bait-prey pairs from the original studies. Why is that so?

Our apologies for the confusion. This threshold was used only for Figure 1D (i.e., only this specific comparison, not for data integration purposes) to provide a fair comparison of the effects on PPI overlap of the matrix models vs. published spoke models.

We did not threshold for the purposes of integrating the datasets. When generating the integrated protein interaction network, we used all protein pairs which were reported by the individual published datasets as well as all protein pairs in which we could generate a hypergeometric test value from AP-MS datasets. (Importantly, we do not consider all matrix protein pairs to be equivalent, as this would generate huge numbers of false positive interactions, but rather every matrix model interaction is assigned a statistical confidence score using our hypergeometric scoring scheme.) The number of protein pairs used in the pipeline, broken down by data source, is listed in Figure 1E under the “Protein Pairs” heading.

We now clarify this point in the legend to Figure 1D by including the text “Sizes of matrix model protein interaction networks were kept constant with published networks only for this analysis while the full networks were used for integration.”

(II.) Yeast-2-hybrid studies are prevalent in PPI studies. Would be useful to incorporate Y2H studies in this pipeline?

We do think Y2H datasets might provide additional value due to their orthogonal nature to the experimental systems used here. We also designed the pipeline to allow for the incorporation of additional orthogonal datasets such as Y2H and aim to include them in the future. However, for the current version, it was important to reserve fully independent protein interaction datasets as tests of our integration, so we opted to hold out the Rolland et al. Y2H dataset as an independent experimental test set (Figure 2E). We now state on p. 8 (1st full paragraph) that “The significant overlap of our complex map with these other datasets also points towards the potential value of integrating the other datasets using the pipeline described here to further improve coverage of the human protein interactome. ”

(III.) In this pipeline, original features derived from different datasets (labs) are used. However, different labs performed MS raw data analysis, as well as the subsequent statistical analysis in different ways. This is especially true for proteomics data where the MS raw data need to go through several stages of analyses and may result in discrepancy in results. Would the results be more accurate and precise if all three MS datasets were re-analyzed from scratch? This should be possible as all proteomics data should be in public depositories.

For each of the datasets the experimental setup was subtly different making a generally applied bioinformatics pipeline problematic. For instance, Bioplex ran replicate samples in a reverse order allowing them to calculate an accurate “entropy” score. Additionally, how replicates are handled across experiments is treated differently. For this work, we opted to rely on the processed results from the three datasets and therefore deferred to the expertise of the labs generating the data to produce informative features.

(IV.) Is there an explanation that the 4 baits co-precipitate all 60 subunits of synaptic bouton complex but they do not pull down each other?

This question gets at a key effect that we believe our matrix model is capable of capturing, that of differential pulldowns of intact complexes by other proteins that interact, whether directly or indirectly, with members of those complexes. We discuss this effect in more detail in our response to Referee #2 below, and illustrate why the matrix model can capture these effects informatively in a new Expanded View Figure EVI.

More specific to the question as to why the baits of this complex do not pull down each other, there are several possibilities including mutually exclusive interactions and nonspecific interactions but we currently do not yet have substantial evidence for any specific model.

Reviewer #2:

The manuscript by Drew et al. attempts to exploit large recent datasets aiming at identifying protein-protein interactions by affinity purification coupled with mass spectrometry (AP-MS) or co-fractionation coupled to mass spectrometry (CF-MS) in order to better define protein complexes.

A better characterization of protein complexes is very much needed, as previous resources (MIPS, CORUM) have not kept pace with high-throughput approaches. The current manuscript attempts to bridge this gap.

The authors reanalyze 3 large datasets (Huttlin et al., Hein et al., Wan et al., all published in 2015), that together encompass "9000 mass spectrometry experiments". They claim higher coverage resulting from this analysis, which in part is due to the re-scoring of the interactions in both the Huttlin and the Hein data, moving away from the "spoke" model used in the initial analysis by the authors. Here, the authors apply a matrix model, scoring prey-prey interactions through a hypergeometric distribution model. Using the top N "interactions" scored in each of the AP-MS datasets with this model as compared to the spoke model used by the initial publications led to a bigger overlap between the Hein and Huttlin datasets.

The authors next combined this rescored AP-MS set with the Wan et al. dataset (to which it should be orthogonal), and used machine learning approaches and clustering of pairwise interactions to identify protein complexes. Comparison to curated complexes in CORUM enabled benchmarking of the different scoring / clustering approaches.

The manuscript itself is well-written, but I have major reservations with the complexes obtained by the approach (from browsing the author's reference site), which are described below.

1. While the matrix expansion certainly will increase the coverage of the datasets, it also introduces issues which are apparently not considered here by the authors. This is particularly apparent when looking at protein complexes for which A binds to either B or B', but never to both. In many instances (on the authors' database), there will be a "complex" that contains A-B-B', which is not biochemically realistic. This is particularly obvious with paralogous proteins (so in theory these could be flagged computationally), but it also happens when B and B' are only related by the interaction motif for A. A spoke expansion model would of course require at least B or B' (and preferably both) to be able to discern the mutually exclusive nature of the interactions, but at least it would have a chance to get at the correct answer. After browsing the website, I am worried that many of the new interactions / new complexes are erroneously called.

Any good interaction map must balance false negative rates and false positive rates. Thus, whether to choose one approach over the other can only be properly evaluated by simultaneously considering the numerical precision and recall of the approaches, as we plot in Figure 2A. When evaluating the spoke model, we observe very high false negatives rates (i.e., very low recall, Figure 2A). We demonstrate an unambiguously positive gain in recall at the same high precision by using our weighted matrix model.

Based on the referee's comments, we now realize there is a potential source of confusion between the traditional matrix model, which does indeed have high false positive rates because all prey-prey interactions are included without consideration of their relative merits, and our weighted matrix model, which specifically addresses this problem. The key difference between the two models, as we have shown in the paper and now additionally illustrate with a new Expanded View Figure EV1, lies in properly weighting the inferred matrix model interactions relative to the observed bait-prey (spoke model) interactions.

Although mutually exclusive interactions are a potential source of false positives in our dataset and are difficult to address using current techniques in the field, we believe our weighted matrix model has the potential to identify mutually exclusive interactions on par if not better than current spoke models. We demonstrate this in the new Expanded View Figure EV1 showing a mutually exclusive interaction between complexes A-B-C-D and A-E. In this example, our hypergeometric test correctly scores the true co-complex interactions (eg. B-C) substantially higher than the false interactions (eg. C-E) due to the accounting for multiple experiments at once (including taking advantage of non-specific interactions). The traditional spoke model does not correctly distinguish these true interactions from the false interactions as demonstrated in Expanded View Figure EV1 in the spoke model's failure to identify the B-C interaction.

The reviewer also suggests two potential strategies to identify mutually exclusive interactions. First, a computational strategy based on paralogs: Although at first glance this would seem like a

potential strategy, we would caution against this approach because it is extremely common to find paralogs legitimately interacting with each other in hetero-oligomeric human protein complexes. Well-established examples supported by extensive structural and biophysical evidence include the 20S proteasome, composed of 4 rings of 7 paralogs each; the CCT chaperonin, a hetero-octamer of paralogs; the Lsm complex, similarly assembled from interacting paralogs; and many other examples exist. Second, the reviewer suggests using reciprocal baits as a way to identify mutually exclusive interactions. This approach certainly has promise and deserves greater attention but due to the ambiguity of AP-MS experiments in determining negative interactions (absence of evidence is not evidence of absence), it is difficult to access if it would improve identification of mutually exclusive interactions. For example, in the reviewer's scenario, if a pulldown of A identifies B and B' and a pulldown of B identifies A only, we cannot be conclusive about whether B' is mutually exclusive with B or whether B' is present but just not identified in the experiment. We can, however, make a statistical argument - and in fact this is precisely what is done with the weighted matrix model approach, as we now graphically show in Expanded View Figure EV1 - for the co-occurrence of two proteins across many experiments and can additionally identify proteins that co-occur less likely than random chance. This statistical argument allows us to appropriately rank pairs of proteins given evidence from the entire dataset and potentially decipher mutually exclusive interactions in a more rigorous manner.

In general, however, identifying mutually exclusive interactions from high throughput experiments remains a difficult problem that could benefit from both new methods to analyze raw data as well as novel clustering algorithms, although we feel our current pipeline has the potential to identify mutually exclusive interactions better than previous methods.

2. The current version of the map should certainly not be called "definitive" (as is currently subtitled on the website), especially as it only integrates data from 3 studies, leaving aside a huge fraction of the protein interaction literature (let alone other issues with the dataset). In particular, several high quality AP-MS studies of more moderate scale could have helped expand the coverage and accuracy of the maps (and perhaps escape issues with matrix expansion). In the discussion, the authors refer to their manuscript as "a framework", which is certainly more appropriate, notwithstanding the issues with the current version of the manuscript.

While we do believe our complex map is currently the most comprehensive to date, this field is fast-moving with more datasets currently being published as well as others yet to be integrated in our pipeline. We have updated our website to reflect the reviewer's concern.

3. This is certainly not the authors' fault, but the "gold standard" that is CORUM does not have "complexes" which are all similarly supported by experimental evidence. Some of the "complexes" are based on a single publication/experiment type (including older AP-MS experiments which may or may not have been performed with appropriate controls). Annotation is also in some cases problematic, and in some cases, the same type of A+B or A+B' situation (even when acknowledged in the original manuscript) is annotated as an A, B, B' complex even in CORUM. This type of situation would, of course, influence the benchmarking of the author's dataset. It is not clear what the step of merging complexes based on the Jaccard coefficient of shared components does to the overall reliability of the CORUM data.

To construct our training and test set, we merged complexes to reduce redundancy, but as the reviewer points out, this could lead to combining mutually exclusive interactions. To address this concern, we determined the number of protein interactions in our positive test set that were not co-complex in the original Corum Core set. We identified 112 pairs out of 15,687 total test interactions (0.7%) that met this criterion and removed them from our test set. All precision recall curves [Figure 2A, 2C, Expanded View Figure EV2A-C] have been recalculated with this new set with no obvious change in performance.

Towards the reviewer's broader question about problems with the Corum dataset, we are aware of the deficiencies of the Corum dataset, which are unfortunately prominent in most literature curated datasets, and no real satisfactory alternatives exist that we know of (balancing our competing requirements of complex coverage and accuracy). We would like to point out that the machine learning tools we apply (e.g., SVM) are robust to small amounts of noise in the training and test sets

and therefore believe that minor problems with the Corum dataset do not impact the overall goal of the project.

4. It is not clear to me how the Hein data, which has triplicates analysis of each bait, was analyzed in relation to the Huttlin data which only analyzed a single bait (did repeated observations influence the weight?).

As briefly mentioned in response to the 1st reviewer's (III) comment, we relied on the original authors' analyses for specific protein interaction features. The Hein et al. published features file did have a small amount of redundancy in terms of protein pairs, in which case we took the mean of the reported metrics. We have updated our methods section to reflect this fact in the 3rd paragraph under the subheading: Calculating protein interaction features from the mass spectrometry datasets.

5. On the website, I realize that the authors are listing the evidence source for the complexes. However, it is unclear what the "hein" and "bioplex" alone sources indicate in the absence of the definition of the bait-prey pairs that have lead to the conclusion that there is indeed a complex being formed.

To improve interpretability of the edgewise evidence sources on the website, we have now added the bait(s) to the evidence labels. We have also added a description of the evidence codes in the form of a help ("??") button. Further, we added cluster maps--similar to ones shown in Figure 6 and Expanded View Figure EV3 of the manuscript--for all complexes under the "images" section of the website. Although such cluster maps only display a subset of the features that were used in our model, we find them useful in interpreting the evidence that was used to predict a given complex, specifically if a complex is supported by multiple independent lines of evidence.

6. The identification of "complexes" by their observations only as "preys" is certainly intriguing. This being said, the authors would need to demonstrate experimentally that somehow proteins identified solely on the basis of their recovery as preys are indeed forming a complex. Many of the proteins in this "synaptic bouton complex" in Figure 3 are associating with different endosomal populations (e.g. all the Rab proteins, and syntaxins) or microtubules (e.g. Map proteins) and while it is not impossible that they indeed form a complex, another possibility is that they simply co-precipitate with sets of baits that are localized to the trafficking machinery and/or the cytoskeleton (can the authors exclude that a membrane-bound organelle/vesicle is co-precipitated?). This needs to be experimentally validated.

Towards the general comment and as touched on above, we have directly measured the false positive rates of interactions determined from our weighted matrix model, and have shown that this model is not more susceptible to false positives as described by the reviewer. A spoke model, due to its focus on only a single AP-MS experiment at a time, would be just as likely (if not more so) to mis-identify non-physical interactions than our weighted matrix model, and additionally commits far more false negative errors. As plotted in Figure 2A as well as Expanded View Figure EV2A-B, we observe substantial gains in prediction performance when incorporating our weighted matrix model features into our protein interaction classifier.

In reference to the more specific comment about the Synaptic Bouton Complex, it is worth noting that the BioPlex AP-MS experimental protocol uses NP40 detergent aimed at disrupting vesicles and although we cannot definitively rule out a membrane-bound organelle/vesicle being co-precipitated under these conditions, we do note that nearly the entire complex was pulled down in four separate experiments, suggesting some degree of specificity in their co-complex interactions. Moreover, out of the 131 human proteins annotated with GO term: Synaptic Vesicle (GO:0008021), only 12 are recovered in the synaptic bouton complex, while a total of 66 are found in complexes in our map. Thus, the synaptic bouton complex has only a subset of the proteins that would be found in a vesicle at the pre-synaptic junction and does not appear to represent the pull-down of an intact synaptic vesicle, or even the subset of one that could potentially be present in a HEK cell. We have updated the manuscript to reflect this analysis (page 8, last paragraph and page 9, 1st paragraph).

Nevertheless, we would like to point out the important role vesicles do have in the function of synaptic transmission and therefore we expect proteins in the Synaptic Bouton Complex to certainly localize to and interact with synaptic vesicles. This intricate relationship of vesicles with the

proteins in the Synaptic Bouton Complex makes this link both interesting and biologically relevant. Regarding the point about interactions with microtubules, the cytoskeleton is also quite important in the process of synaptic transmission and we suspect the cytoskeleton is aiding in the arrangement of proteins in the complex. Although the Synaptic Bouton Complex may extend the traditional definition of a protein complex, we believe the complex cleanly falls into our definition of a complex being a physically associated set of proteins that carry out a specific function, primarily that of synaptic transmission.

7. I was not clear from the description of Figures 4-6 whether an attempt was made at directly comparing the authors' network with those of Boldt and Gupta who performed large scale directed proteomics studies to profile the centrosome and cilium (e.g. CCDC138 was uncovered - though not validated - in the Gupta paper).

We have now updated Figure 4 to include a direct comparison of the Boldt and Gupta datasets in terms of coverage of SysCilia and ciliopathy proteins. Our map exceeds coverage in all categories when compared directly to other complex maps, including the Boldt map. Additionally, when compared to the specific categories of ciliopathy genes and SysCilia, our map is on par with and exceeds, respectively, the targeted protein interaction networks of Boldt and Gupta. It's worth restating that our map, however, is proteome-wide, and we thus expect this high level of performance to be generally true across many specific disease types.

We have updated the last paragraph on page 9 and 1st full paragraph on page 10 of the manuscript to address this analysis.

8. On Figures 5 to 7, it would have been nice to show the evidence source on the edges to better highlight the importance of the new dataset in comparison to the individual datasets in generating new knowledge. For example, it seems (looking at the individual datasets) that all the raw data about WDR90 comes from Hein et al. (the same seems to be true for KIAA1328 while the converse - all hits from Huttlin - exist for ANKRD55). The authors should make a better case for the specific need for hu.MAP to make these hypotheses.

With regard to the question of showing specific lines of evidence, as noted above in the response to comment #5, we have updated the supporting web site to now include the source of evidence for every interaction in the map, including the type of supporting evidence (e.g. bait-prey, prey-prey, fractionation evidence) as well as specific baits used for AP-MS experiments.

More generally, the question of what is the specific need for hu.MAP is an important one and one that we definitely want to address clearly. hu.MAP advances the state of the art in three fundamental ways:

First, we find additional signal (i.e., previously unidentified protein interactions) in AP-MS datasets using our weighted matrix model approach. The impact of this advance is clearly evident in our highlight of the Synaptic Bouton Complex which was based almost entirely on weighted matrix model edges. Weighted matrix model edges also predominate in the vast majority of complexes in the map including the centriolar complex described in Figure 6. Specifically, 14 out of 17 edges involving WDR90 are exclusively identified by our weighted matrix model approach including many that have a confidence score greater than those of the bait-prey interactions. Additionally, 15 out of 17 of edges involving KIAA1328 are exclusively identified by our weighted matrix model approach including the majority (11) having a confidence score greater than bait-prey interactions. These complexes would not have been identified from the individual datasets without the weighted matrix model.

Second, the integration of multiple datasets from various experimental setups in a machine learning framework allows for the compounding of evidence for a given protein interaction. This not only has an impact on the edges that have support from multiple sources but also, by way of a more refined scoring and ranking system, edges that have support from only a single source (i.e., we can better distinguish true from false positives within individual datasets). The impact of this can be seen in Figure 2D, where >10k protein interactions are supported by multiple lines of evidence and are enriched for highly confident scores.

Third, the clustering procedure we employ, which optimizes clustering parameters based on gold standard training complexes, allows us to better remove false positives (e.g., as in Figure 2C), as well as to clarify the organization of proteins into physically associated functional modules that are otherwise hidden in a difficult to interpret “hairball” protein interaction network. The inclusion of ANKRD55 as a member of the IFT-B complex is an excellent example of the utility and specificity of our clustering procedures over previous work. The evidence for ANKRD55’s association with IFT-B is entirely based on data from BioPlex, but ANKRD55 fails to appear in the BioPlex complex map. This observation, along with our performance evaluation of complex maps on an independent set of test complexes in Expanded View Figure EV2D-E, suggests that our clustering procedure can identify complexes otherwise missed by traditional clustering techniques.

It is worth noting that since all of the complexes in our map are derived from published datasets, one will often be able to retrospectively go back to the original publications and find support for a given complex there. We feel this argument ignores the importance of properly scoring and ranking protein interactions, as well as clustering interactions into high confident predicted complexes. As mentioned above in response to comment 1, the value of these approaches can only be rigorously evaluated in a quantitative recall/precision framework, in which our map shows substantial gains.

We thank the referee for raising this critical point and have now clarified the discussion section on page 12-13 of the manuscript to address this comment as well as the last paragraph of page 11.

Minor comments

1- It would be nice to present legends for the tables (as well as more explicit titles)

We have added legends as a separate sheet in the excel spreadsheets and have updated the titles to be more descriptive.

2- Pictures of ciliopathies from Figure 5 are not relevant as presented; remove, unless there is some particular point that needs to be made (and which is currently not explained well).

Figure 5 has two main objectives. First it demonstrates the direct link between disrupted complexes and clinical presentations in human patients showing the great relevance our map has in relation to human disease. Second, it demonstrates our ability to make strong predictions as to the disease association of uncharacterized proteins due to their physical association to other known disease genes. For example, in the manuscript we describe ANKRD55 as a new member of the IFTB machinery based on its co-occurrence with other IFTB subunits. Many IFTB subunits have similar disease phenotype associations (ex. chest narrowing, polydactyly) and therefore we make the prediction that patients with disruptions in ANKRD55 will have similar clinical presentations. Since our complex map includes a large number of uncharacterized proteins that are co-complex with genes that have known disease phenotypes, our map is likely to be a powerful tool for clinicians looking for mutated proteins in patients who do not yet have known molecular causes to their phenotypes.

We have updated figure 5 and the text on page 10, 2nd full paragraph, to clarify this point.

3- Since the same type of benchmarking was also used in the selection of the scoring for Wan et al. (as far as I can tell), the authors should probably expand on how they avoid circularity in the scoring for those interactions which are supported by the CF-MS data.

To construct our machine learning classifier, all features used from the three datasets were calculated from raw data and no input feature was trained using instances from our benchmark. More explicitly, all three published datasets generated raw data from experiments (ex. AP-MS, CF-MS), calculated features from these raw data (ex. zscore, correlation coefficients) and then used to train a model to build a set of high confident protein interactions. We do not use the “post-trained” set of high confident protein interactions from the published sets as input into our model but rather use the “pre-trained” calculated features. Since these features are “pre-trained” and come directly from the raw experimental data there is no circularity with the benchmark. We have updated the text to clear up this point of potential confusion in the first paragraph of page 6 and page 14, 3rd full paragraph.

Reviewer #3:

This manuscript describes an effort to assemble a catalogue of human protein complexes by integrating data from different sources, including a large number of mass spectrometry based measurements that are indicative of protein protein interaction. Drew and colleagues have compiled features from 3 different recent studies and integrated these using an SVM based approach. The ranked list of protein interactions were then clustered into protein complexes. The pairwise interactions and derived protein complexes were properly benchmarked against unseen data. This updated view of human interactions and complexes was then used to study in more detail ciliary proteins, identifying novel components of ciliary protein complexes that were validated by localization. A novel member of the IFT complex was also discovered and validated experimentally.

This work is most related with previous efforts from the same group at consolidating and integrating evidences for human protein interactions (e.g. Lee et al. Genome Research 2011). This is a very timely effort since so much new information has come out on human protein interactions and each individual study has been kept isolated. There is a great opportunity here to provide an updated human interaction/complex version that is of much higher coverage and accuracy than other previous attempts. This group has a lot of past experience in these approaches and the work is well done. However, I have some concerns about some of the technical aspects, in particular on not being sufficiently explicit about the use of some of the data sources such as the yeast, work and fly data or the human co-expression information. From a presentation point of view the article could also focus less of some of the computational details and perhaps more on the delivered interactions and complexes resource.

Major concern:

1- Superficially the authors appear to integrate 3 mass spectrometry based studies as stated throughout the manuscript. However, in the methods section one learns that the features used from the Wan et al. study include 19 features from HumanNet that include a very diverse set of data types and orthology transferred data. For example these include the large survey of genetic interactions from yeast, human co-expression information, human co-IP and human yeast-two-hybrid data, among several others. This is very misleading. I assume that the authors did not make this explicit in the manuscript in order to simplify the narrative but I think it is important to inform the readers more clearly about this. In particular, it may give an incorrect sense of relative value of the co-fractionation experiments to the readers.

Thank you for pointing out this potentially confusing description of our methods. HumanNet features were only included for the pairs of proteins for which there was evidence in the fractionation experiments and had entries in HumanNet. Only 35,028 protein pairs out of the 2,557,860 total pairs have evidence from HumanNet (1.3%). We have clarified our methods to reflect this fact (4th full paragraph on page 14) and address other concerns below, including confirming the (very minor) effect of omitting HumanNet features altogether.

1.1 - The authors should consider the co-fractionating features separately in their study. In Figure 1E, this would mean clearly showing that some features are transferred by orthology and include additional data types such as gene expression, Y2H, genetic-interactions, etc. In Figure 2A it is again important to benchmark separately the features from co-fractionation from the HumanNet features from the co-IP data from the other 2 studies.

We agree that considering fractionation-only evidence is informative to the discussion and have added a precision recall curve to figure 2A calculated using only co-fractionation data. We also include additional precision recall curves in Extended View Figure EV2C discussed below. In regards to Figure 1E, as discussed above, extra evidence was only included for protein pairs that had fractionation evidence.

1.2 - From the HumanNet set of features, the authors should not consider any feature that is curated from the literature or based on co-citation even for species other than human. Stable protein complexes tend to be highly conserved across species so literature curated complexes from other species should be avoided in the training. This includes at least the features SC-LC, SC-CC, CE-LC and CE-CC.

*Although Corum is mammalian-specific and comprises only complexes identified in primary literature from those species to be included in the set, we do recognize the slim potential for circularity with evidence for complexes found in yeast and *C. elegans* and agree with the referee that this point deserved further investigation. We therefore re-trained our classifier without features from SC-LC, SC-CC, CE-LC and CE-CC and evaluated using our leave-out test set. The results are now included in Expanded View Figure EV2C and demonstrate no difference in performance when these features are used. As noted earlier, we additionally trained a classifier without any HumanNet features and observed negligible performance loss Expanded View Figure EV2C. As mentioned above, only a small fraction of protein pairs had evidence from HumanNet and therefore had only minimal impact on the final performance.*

These observations point to a broader conclusion that animal mass spectrometry protein interaction datasets have reached a sufficient point where adding in supporting non-physical interaction information (from expression, etc) is no longer necessary to support protein interaction discovery. We now note this on in the 2nd full paragraph on page 6 of the manuscript.

1.3 - There are a large number of protein interactions in human derived from Y2H experiments. Together with the 3 works integrated in this study the Y2H networks from Vidal and colleagues would also be one very large set of interactions that has been published recently. Some of the Y2H interactions are included as part of the HumaNet features, which as stated above, I find misleading. The most recent large Y2H study was used by the authors to validate the predictions. Why haven't the authors consider them all as a feature ? Was it again mostly to simplify the narrative of the manuscript or did they not provide additional predictive power ?

As mentioned in response to Reviewer 1, comment (II), we agree that incorporation of Y2H data could provide additional gains, but we felt it was more valuable to utilize Y2H as an orthogonal dataset for comparison purposes.

1.4 - How up-to-date are the HumanNet features ? Are these just taken directly as were done for Lee et al. ?

The features used were identical to those in Lee et al. (and Wan et al.) We note this on p. 14, paragraph 4.

Minor concerns

1 - How were the missing values handled for the SVM ? The feature matrix is certainly extremely sparse. Were all the missing values treated similarly ? In some cases the absence of interaction could be interpreted as a negative information but this will not be equally so for all methods and all interactions.

All missing values were assigned values of 0.0 in the raw feature matrix. We have updated the methods section on page 15 to reflect this.

2 - The title is somewhat of a gimmick. The network integrates other experiments beyond MS data and the fact that there were 9000 MS "experiments" integrated is itself not the result. The authors are using genetic interaction data from yeast that include now around 539,710 interactions according to biogrid or gene expression data for human that probably also includes a very high number of samples. I would suggest to remove the reference to the mass spectrometry data from the title. Perhaps focusing on the comprehensiveness of the derived interaction network and core complexes.

We wish to push back (gently) against this criticism in light of our explanations above. As mentioned above in response to comment 1, any genetic interaction data or expression data used was in concert with fractionation data (and never without it) and in fact provided only a minimal amount of overall raw data as well as negligible performance gains. Thus, virtually all of the relevant data in the map stems directly from mass spectrometry proteomics experiments, and the title reflects this. We feel that stating the number of MS experiments is highly useful to the reader to illustrate the sheer amount of data that supports complexes in our map.

3 - Some of the benchmarking sections could be described more succinctly in the text. In particular the evaluation of protein complexes using the k-cliques strategy. It is very important to properly

benchmark the interactions and complexes and I think this has been well done in this work. However, for a wider audience this section could be reduced.

As this is the first description of the k-cliques method, we felt a full derivation was necessary. However, in response to this comment, we have now moved the derivation of k-cliques into a supplemental methods section.

4 - Having a look at the list of predicted complexes it looks like many complexes have just 2 subunits. It could be useful for readers to have a figure showing the number of complexes having N number of subunits.

We now include a histogram of complex sizes as Extended View Figure EV2F and have updated the text on page 7, last paragraph.

5 - In figure 5, do the authors have permission to use the phenotype figures from the other articles ?

We have gained permission to reuse the images in figure 5 and they are properly cited.

2nd Editorial Decision

18 April 2017

Thank you for sending us your revised manuscript. We have now heard back from the reviewer who agreed to evaluate your study. As you will see below, this reviewer thinks that the previously raised concerns have been satisfactorily addressed. S/he lists however a few remaining minor issues, which we would ask you to address in a revision.

REFEREE REPORT

Reviewer #3:

The authors have addressed the concerns I had brought up in the previous report. In particular they have shown that the other types of evidences not based on mass-spectrometry are almost negligible in their predictor. It is still informative to know that this is the case so I won't suggest to remove this aspect from their work.

I still don't like the title too much but it is subjective and the authors like the emphasis on the number of MS experiments that were pooled into the predictor.

While reading the article again I thought that the fact that some complexes have not enrichment in term annotations (GO, Reactome, KEGG, etc) could just be because so many of them have 2 or 3 subunits. There are unlikely to result in strong enrichment just due to the small size. If this is the case the authors could add a sentence saying the fraction of complexes with 3 or more sub units that have at least 1 annotation.

As I had suggested in the previous report, the authors added an histogram with the number of subunits per complex. I think it be worth also mentioning in the results section the fraction of complexes with 2 subunits. Some readers may, as I did, incorrectly think that complexes refers to larger assemblies.

When opening the supplementary table with the list of complexes with libreoffice (in Ubuntu) or in Excel (also in Ubuntu) there was a non standard character separating the ID and gene names. It opens well in Excel for Mac.

I think this is will be a great resource with many applications. Several analysis rely on a high confidence list of protein complexes and this is much improved version relative to the often used CORUM complexes.

Responses to referee

Reviewer #3:

The authors have addressed the concerns I had brought up in the previous report. In particular they have shown that the other types of evidences not based on mass-spectrometry are almost negligible in their predictor. It is still informative to know that this is the case so I won't suggest to remove this aspect from their work.

We agree the topic is informative and thank the reviewer again for bringing up the point.

I still don't like the title too much but it is subjective and the authors like the emphasis on the number of MS experiments that were pooled into the predictor.

We have updated the title in light of the editor's suggested title.

While reading the article again I thought that the fact that some complexes have not enrichment in term annotations (GO, Reactome, KEGG, etc.) could just be because so many of them have 2 or 3 subunits. There are unlikely to result in strong enrichment just due to the small size. If this is the case the authors could add a sentence saying the fraction of complexes with 3 or more sub units that have at least 1 annotation.

*This is a good point and one which we took the opportunity to explore further and clarify in our updated manuscript. Nearly all of the complexes in hu.MAP (4646/4659) have at least one significantly enriched annotation ($p\text{-value} \leq 0.05$) after FDR (Benjamini-Hochberg) multiple hypothesis correction when searched individually, but this does not take into account the multiple testing across the $>4,000$ total complexes. To directly account for this effect, we permuted the assignment of proteins to complexes across hu.MAP (shuffling the proteins across complexes while retaining the size distribution of the complexes) and tested for significantly enriched annotations on the shuffled complexes. The new **Figure EV3** shows the distribution of the largest $-\log(p\text{-values})$ (i.e., most significant annotation) for each complex for both hu.MAP and the shuffled complexes. **Figure 4B** reports the set of hu.MAP complexes with a significantly enriched annotation at a false discovery rate of 5% with respect to the shuffled set of complexes. Greater than 40% (1,880 out of 4,659) of the complexes had at least one significantly enriched annotation term, demonstrating the biological pertinence of complexes in the map, well in excess of shuffled complexes of the same sizes.*

To directly address the reviewer's comment regarding complex sizes, we next examined the size distribution of these 1,880 complexes relative to the complexes which did not show strong functional enrichment. As expected by the referee, the non-enriched complexes did tend to be smaller, although it is worth noting that many complexes of size 2 and 3 were included in the 1,880 enriched set. In contrast, larger complexes were increasingly more likely to show functional enrichment. Overall, 1,514 of the enriched complexes contained 3 or more subunits. We now indicate this trend in the text on p. 9.

As I had suggested in the previous report, the authors added an histogram with the number of subunits per complex. I think it be worth also mentioning in the results section the fraction of complexes with 2 subunits. Some readers may, as I did, incorrectly think that complexes refers to larger assemblies.

We have updated the text in the last paragraph on page 7 to explicitly distinguish complexes with 2 or greater than 2 subunits.

When opening the supplementary table with the list of complexes with libreoffice (in Ubuntu) or in Excel (also in Ubuntu) there was a non standard character separating the ID and gene names. It opens well in Excel for Mac.

We have updated the supplemental tables to fix this problem.

I think this is will be a great resource with many applications. Several analysis rely on a high confidence list of protein complexes and this is much improved version relative to the often used CORUM complexes.

We thank the referee for their appreciation of this work and the time they invested in reviewing it.

3rd Editorial Decision

12 May 2017

Thank you again for sending us your revised manuscript. We are now satisfied with the modifications made and I am pleased to inform you that your paper has been accepted for publication.

Corresponding Author Name: Edward M. Marcotte

Manuscript Number: MSB-16-7490